# How Distributed Collaboration Influences the Diffusion Model Training? A Theoretical Perspective

Jing Qiao [1 2]   Yu Liu [1]   Yuan Yuan [3 4]   Xiao Zhang [1]   Zhipeng Cai [5]   Dongxiao Yu [1]

## Abstract

This paper examines the theoretical performance of distributed diffusion models in environments where computational resources and data availability vary significantly among workers. Traditional models centered on single-worker scenarios fall short in such distributed settings, particularly when some workers are resource-constrained. This discrepancy in resources and data diversity challenges the assumption of accurate score function estimation foundational to single-worker models. We establish the inaugural generation error bound for distributed diffusion models in resource-limited settings, establishing a linear relationship with the data dimension $d$ and consistency with established single-worker results. Our analysis highlights the critical role of hyperparameter selection in influencing the training dynamics, which are key to the performance of model generation. This study provides a streamlined theoretical approach to optimizing distributed diffusion models, paving the way for future research in this area.

## 1. Introduction

Diffusion models have significantly challenged the prominence of Generative Adversarial Networks (GANs) (Goodfellow et al., 2014) and have become a central focus in the realm of data generation (Yang et al., 2023; Song & Ermon, 2019; Dhariwal & Nichol, 2021). These models operate by introducing noise to data through a forward process and subsequently learning to reverse this perturbation to generate new samples (Yang et al., 2023; Song et al., 2020), leading to notable advancements across fields like computer vision (Harvey et al., 2022; Saharia et al., 2022), natural language processing (Austin et al., 2021; Li et al., 2022), and temporal data modeling (Lopez Alcaraz & Strodthoff, 2023; Tashiro et al., 2021).

Theoretical research on diffusion models has traditionally been framed within the single-worker paradigm, yielding substantial progress in defining polynomial error bounds. These studies generally use measures such as total variation distance or KL divergence to quantify the discrepancies between true and approximate paths. Notable advancements include using Girsanov's methodology to achieve error bounds that scale linearly with data dimensions under early-stopping conditions (Benton et al., 2024) and adopting stochastic control perspectives to derive comparable bounds under specific smoothness conditions (Conforti et al., 2023). These insights significantly aid single-worker training of diffusion models.

However, the trend towards geographically dispersed data sources makes centralizing data for processing not only costly but also privacy-invasive, prompting a shift towards distributed diffusion model training (Tun et al., 2023). Despite this practical shift, theoretical exploration remains largely anchored in the single-worker setup, leaving the impacts of distributed training on diffusion models largely unexplored and not well understood (Li et al., 2024).

To effectively develop theoretical insights for distributed diffusion models, several critical questions must be addressed: **(1) How to facilitate distributed collaboration in diffusion model training?** Privacy preservation is paramount, as diffusion models often handle sensitive data like sound and images. Traditional methods that distribute fragments of data for collaborative training do not adequately safeguard privacy, rendering them unsuitable (Li et al., 2024). Furthermore, the resource variance among workers complicates participation; low-resource workers may struggle with full model training, potentially leading to an unfair model. Simultaneously, accommodating slower workers could introduce delays, known as the straggler problem, thus hindering overall training efficiency. Balancing inclu-

---
[1]School of Computer Science and Technology, Shandong University, Qingdao, China [2]Zhongtai Securities Institute for Financial Studies, Shandong University, Jinan, China [3]School of Software, Shandong University, Jinan, China [4]Joint SDU-NTU Centre for Artificial Intelligence Research (C-FAIR), Shandong University, Jinan, China [5]Department of Computer Science, Georgia State University, Atlanta, GA, USA. Correspondence to: Dongxiao Yu <dxyu@sdu.edu.cn>.

*Proceedings of the 42$^{nd}$ International Conference on Machine Learning*, Vancouver, Canada. PMLR 267, 2025. Copyright 2025 by the author(s).

sivity and efficiency is therefore essential for successful distributed collaboration. **(2) How to evaluate the generation quality of collaboratively trained diffusion models?** The collaboration of multiple workers, each with different resource constraints and data heterogeneity, introduces unique challenges in assessing the theoretical performance of diffusion models. Limited resources might force some workers to engage in sparse training, potentially accumulating errors that degrade model performance. Moreover, variations in the original data distributions among workers can cause inconsistencies in the quality of data generated by the unified model, complicating the application of traditional single-worker evaluation metrics like accurate score function estimation (Benton et al., 2024; Chen et al., 2023b;a). **(3) Can appropriate hyperparameters be selected to optimize generation quality?** Certain hyperparameters, including learning rate and noise scheduling, critically influence the generation quality of diffusion models. For example, noise scheduling plays a pivotal role in the denoising phase, directly affecting the quality and diversity of the generated outputs. Therefore, identifying and adjusting these hyperparameters is crucial to maximize the efficiency and effectiveness of the diffusion model.

In this paper, we present the first theoretical performance evaluation for distributed diffusion models under resource-constrained scenarios. Initially, to maintain data privacy, each worker independently processes the noising and denoising stages using locally scheduled discrete-time noise, learning a parameterized score function that is shared with a central server. Specially, we allow for sparse training tailored to each worker's available resources, enhancing computational efficiency significantly. To evaluate the generation quality of the models, we utilize Girsanov's theorem to link discrepancies between ideal and actual data distributions to factors such as time discretization, distributed training dynamics, and equivalent loss substitution. For distributed training dynamics, we manage sparse training errors through coordinate-wise model aggregation and bound the impact of data distribution heterogeneity with auxiliary functions. Additionally, we calculate the expected deviation in the drift terms caused by time discretization and demonstrate that score matching during the denoising phase incurs a consistent numerical error. Lastly, by carefully adjusting hyperparameters, including the learning rate and the intervals of discrete time, we ensure that the generation error bound is predominantly dictated by the dynamics of distributed training, thereby elevating the overall efficacy of the diffusion models.

The core contributions can be summarized as follows:

- We propose an innovative training mechanism for distributed diffusion models that prioritizes data privacy by keeping private data localized through synchronized noise scheduling, while adapting to varying resource constraints via local sparse training.

- To the best of our knowledge, this is the first time a generation error bound has been established for distributed diffusion models under resource limitations, which matches the best-known results of the single-worker paradigm (Benton et al., 2024) and shows a linear relationship with the data dimension $d$.

- We detail strategic hyperparameter adjustments, such as noise scheduling and learning rate optimization, to ensure that the generation error bound is chiefly influenced by the dynamics of distributed training, thus enhancing overall model performance.

## 2. Related Work

Diffusion models have recently risen to prominence due to their exceptional performance in various domains such as computer vision (Harvey et al., 2022), natural language processing (Li et al., 2022), and multi-modal learning (Ramesh et al., 2022), challenging the once-dominant Generative Adversarial Networks (GANs) in terms of both stability and efficiency of generation (Dhariwal & Nichol, 2021).

Theoretical investigations into diffusion models have deepened our understanding of their mechanics, particularly regarding convergence rates, stability, and data generation quality. Initial theoretical explorations required stringent assumptions about data distributions, such as compliance with log-Sobolev inequalities (Yang & Wibisono, 2022), and often resulted in either non-quantitative bounds (Pidstrigach, 2022) or exponential dependency on problem parameters (Block et al., 2020). Recent advancements have mitigated these limitations, with studies like those by Chen et al. (2023b) achieving polynomial error bounds in total variation distance without restrictive data distribution assumptions, using the Girsanov change of measure to quantify discrepancies between true and approximated reverse processes. Further progress by Chen et al. (2023a) refines this approach by expanding the Girsanov methodology and introducing pivotal theorems that address the behavior of KL divergence in relation to data dimensions and the impacts of early-stopping on error magnitudes. Moreover, Benton et al. (2024) have significantly advanced this line of work by establishing state-of-the-art error bounds that are linear in the data dimension, even in the absence of smoothness assumptions on the data distribution.

Despite these theoretical advances, the focus remains predominantly on single-worker setups. The shift towards distributed training, as seen in works like those by Zhou et al. (2024) and Lian et al. (2017), highlights the evolving landscape in response to the scalability demands of big data. Tun et al. (2023) explore the FL strategy to train diffusion

models, paving the way for the development of federated diffusion models. Stanley Jothiraj & Mashhadi (2024) introduce Phoenix that integrates various strategies to enhance the diversity of generated samples, even when the training data exhibit statistical heterogeneity. DistriFusion (Li et al., 2024) exemplifies this trend, facilitating the parallel processing of diffusion models across multiple devices to reduce latency in sample generation without sacrificing quality.

However, a theoretical understanding of how distributed diffusion models perform under resource constraints remains underexplored, indicating a crucial gap in the literature that future research needs to address. This presents an opportunity to extend the robust theoretical groundwork laid by single-worker studies to more complex, distributed environments, potentially unlocking new efficiencies and capabilities in diffusion model applications.

## 3. Training Description

In our framework for training distributed diffusion models, we employ a setup that includes $N$ workers and a central server. For each worker-$n$ ($n = 1, \cdots, N$), the initial phase is to progressively transform the given data distribution $q_{n,0}$, into a known prior distribution. This is referred to as the forward process, and it can be described using the Ornstein-Uhlenbeck (OU) process via the stochastic differential equation (SDE) (Pedrotti et al., 2024):

$$\mathrm{d}X_{n,t} = -X_{n,t}\mathrm{d}t + \sqrt{2}\mathrm{d}B_{n,t}, \quad X_{n,0} \sim q_{n,0} \quad (1)$$

where $(B_{n,t})_{t \in [0,T]}$ denotes a standard Brownian motion on $\mathbb{R}^d$. Equation (1) aligns with a methodology known as Denoising Diffusion Probabilistic Models (DDPMs) (Ho et al., 2020), and is also referred to as Variance Preserving SDE in (Song et al., 2020). The OU process is favored for its analytically tractable transition densities, and it holds that $X_{n,t}|X_{n,0} \sim \mathcal{N}(X_{n,0}e^{-t}, (1 - e^{-2t})\boldsymbol{I}_d)$.

We use $q_{n,t}(X_{n,t}), t \in [0,T]$ to denote the marginals of the forward process for each worker-$n$ ($n = 1, \cdots, N$) and then the ideal reverse process satisfies the SDE:

$$\begin{cases} \mathrm{d}X_{n,t} = -\{X_{n,t} + 2\nabla \log q_{n,t}(X_{n,t})\}\mathrm{d}t + \sqrt{2}\mathrm{d}\tilde{B}_{n,t} \\ X_{n,0} \sim q_{n,0} \end{cases}$$
$$(2)$$

where $(\tilde{B}_{n,t})_{t \in [0,T]}$ is another standard Brownian motion on $\mathbb{R}^d$. Aligning with the idea of reconstructing the data distribution from noise, the reverse process (2) can be transformed to a forward one by inverting the time direction $t$ with $T - t$ and setting $X_{n,t} = Y_{n,T-t}$:

$$\begin{cases} \mathrm{d}Y_{n,t} = \{Y_{n,t} + 2\nabla \log q_{n,T-t}(Y_{n,t})\}\mathrm{d}t + \sqrt{2}\mathrm{d}B'_{n,t} \\ Y_{n,0} \sim q_{n,T} \end{cases}$$
$$(3)$$

where $(B'_{n,t})_{t \in [0,T]}$ is the standard Brownian motion on $\mathbb{R}^d$. Ideally, the process $(Y_{n,t})_{t \in [0,T]}$ can thus generate samples from the distribution $q_{n,0}$ by sampling $Y_{n,0} \sim q_{n,T}$.

For each worker-$n$ ($n = 1, \cdots, N$), the score function $\nabla \log q_{n,T-t}(Y_{n,t})$ described in equation (3) is not directly accessible, necessitating the use of an estimated function $s_\theta(Y_{n,t}, T - t)$ to approximate $\nabla \log q_{n,T-t}(Y_{n,t})$ throughout the interval $t \in [0, T]$.

Given that equation (3) outlines a continuous-time process, practical implementation requires discretization of the time variable. This is achieved by segmenting the continuous timeline into a series of discrete intervals $0 = t_0 < t_1 < t_2 < \cdots < t_K \leq T$. The process begins with sampling $Y_{n,0}$ from the distribution $q_{n,T}$, followed by sequentially solving the SDE, also referred to as the exponential integrator (Zhang & Chen, 2023; Bortoli, 2022; Chen et al., 2023a), for each interval $[t_k, t_{k+1}]$ where $k = 0, \cdots, K - 1$.:

$$\mathrm{d}Y_{n,t} = \{Y_{n,t} + 2s_\theta(Y_{n,t_k}, T - t_k)\}\mathrm{d}t + \sqrt{2}\mathrm{d}\hat{B}_{n,t}$$

where $(\hat{B}_{n,t})_{t \in [0,T]}$ is a standard Brownian motion. We denote the length of the $k$-th discretized time interval by $\gamma_k = t_{k+1} - t_k$.

Considering the potential constraints on the quality and quantity of training data available to each worker, our objective is to leverage the collective capabilities of $N$ workers to collaboratively train a score function estimator $s_\theta(\cdot)$. Typically parameterized as a neural network, this function is defined by a parameter vector $\theta \in \mathbb{R}^D$. Our primary goal is to optimize the performance of this model by minimizing the following loss function:

$$\frac{1}{N}\sum_{n=1}^{N} \mathcal{L}_n(\theta) \quad (4)$$

where the local objective $\mathcal{L}_n(\theta)$ is denoted as $\mathcal{L}_n(\theta) = \sum_{k=0}^{K-1} \gamma_k \mathbb{E}[\| \nabla \log q_{n,T-t_k}(Y_{n,t_k}) - s_\theta(Y_{n,t_k}, T - t_k) \|^2]$.

To address the unavailability of the score function $\nabla \log q_{n,T-t_k}(Y_{n,t_k})$, we replace $\mathcal{L}_n(\theta)$ with an equivalent objective $F_n(\theta)$ using denoising score matching (Vincent, 2011). The equivalence of the two is given by Lemma 4.8. And $F_n(\theta)$ is expressed as follows:

$$\sum_{k=0}^{K-1} \gamma_k \mathbb{E}[\| s_\theta(Y_{n,t_k}, T - t_k) - \nabla \log q(Y_{n,t_k}|X_{n,0}) \|^2]$$
$$(5)$$

Due to the inherent randomness in sampling during training, we introduce a stochastic local loss function, denoted as $f_n(\theta, \xi_n)$. Specifically, we assume that $f_n(\theta, \xi_n)$ is unbiased, which is a common assumption in distributed settings (Lian et al., 2017), meaning that $\mathbb{E}[f_n(\theta, \xi_n)] = F_n(\theta)$.

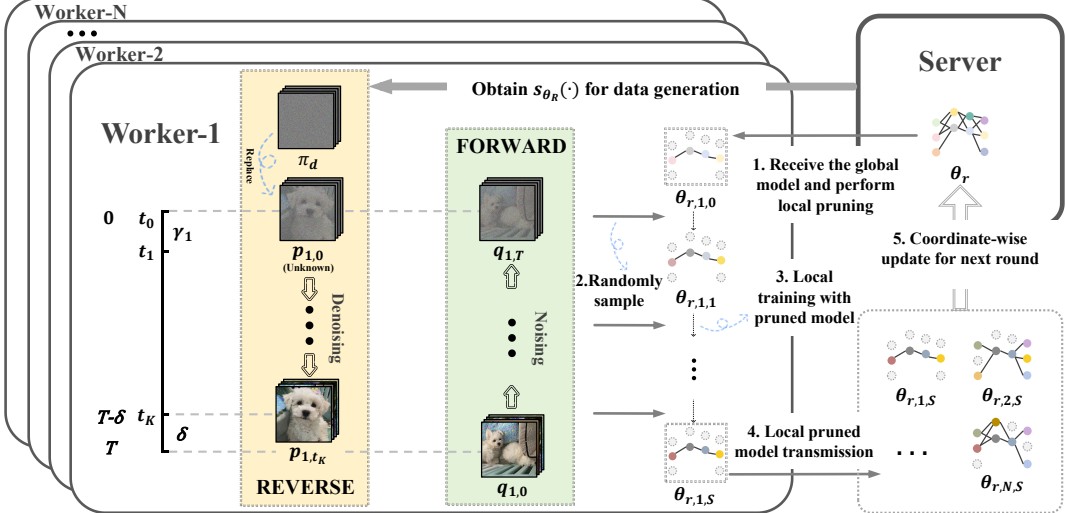

*Figure 1.* The illustration of distributed diffusion model training with pruning.

With this formulation, our objective can be rewritten as the following optimization problem:

$$\min_{\theta \in \mathbb{R}^D} F(\theta) := \frac{1}{N} \sum_{n=1}^{N} \underbrace{\mathbb{E}[f_n(\theta, \xi_n)]}_{:=F_n(\theta)} \quad (6)$$

Given the potential constraints and variability in available resources across workers, we enable each worker to train only a subset of the full model parameters. More specifically, the learning process in round $r$ for achieving (6) can be described in detail as follows:

- **Mask Generation and Model Pruning:** Each worker-$n$ generates a mask $m_{r,n} \in \{0,1\}^D$ based on its own resources. Once worker-$n$ receives the latest global model parameter $\theta_r \in \mathbb{R}^D$ from the server, it performs pruning operations based on its local mask, resulting in the initial pruned model parameters $\theta_{r,n,0} = \theta_r \odot m_{r,n}$ in this round $r$.

- **Local Training with Pruned Model:** Each worker-$n$ performs $S$ steps of local training on the pruned model. The model parameters are updated in each step according to the gradient of the local loss $\nabla f_n$ and the local mask $m_{r,n}$:

$$\theta_{r,n,s} = \theta_{r,n,s-1} - \eta \nabla f_n(\theta_{r,n,s-1}, \xi_{n,s-1}) \odot m_{r,n} \quad (7)$$

Here, $s$ represents the current step, $\theta_{r,n,s-1}$ is the model parameter from the previous step, and $\eta$ is the local learning rate.

- **Model Update and Synchronization:** After $S$ steps of local training, each worker-$n$ obtains the final pruned model parameters $\theta_{r,n,S}$ in the round $r$. These parameters are then sent back to the server to update and synchronize the global model. For each coordinate $i = 1, 2, \cdots, D$, the parameter is updated as:

$$\theta_{r+1}^{(i)} = \frac{1}{|N_r^{(i)}|} \sum_{n \in N_r^{(i)}} \theta_{r,n,S}^{(i)} \quad (8)$$

where $N_r^{(i)} = \{n : m_{r,n}^i = 1\}$ and we denote $\Gamma^* = \min_{r,i} |N_r^{(i)}| \geq 1$.

After completing $R$ rounds of distributed training (each with $S$ steps), we obtain the score function estimation $s_{\theta_R}(\cdot)$. We can then solve the following process to approximate the initial data distribution $q_{n,0}$:

$$\begin{cases} \mathrm{d}Y_{n,t} = \{Y_{n,t} + 2s_{\theta_R}(Y_{n,t_k}, T - t_k)\}\mathrm{d}t + \sqrt{2}\mathrm{d}\hat{B}_{n,t}, \\ \qquad\qquad\qquad\qquad\qquad\qquad t \in [t_k, t_{k+1}] \\ Y_{n,0} \sim q_{n,T} \end{cases} \quad (9)$$

Since the initial distribution $q_{n,T}$ is not directly accessible, we instead sample $Y_{n,0}$ from the standard Gaussian distribution $\pi_d$ leveraging the fact that the Ornstein-Uhlenbeck (OU) process converges exponentially fast to the standard Gaussian (Bakry et al., 2014; Chen et al., 2023a).

Furthermore, rather than running (9) to approximate the initial data distribution $q_{n,0}$ we opt to approximate $q_{n,\delta}$ as an early-stopping measure (Song et al., 2020), setting $t_K = T - \delta$. This approach is justified by the fact that for a sufficiently small $\delta$, the difference between $q_{n,0}$ and $q_{n,\delta}$

is negligible. Additionally, this strategy mitigates potential issues where $\nabla \log q_{n,t}$ may grow rapidly or even "explode" as $t$ approaches zero, particularly in cases involving non-smooth data distributions.

As a result, starting from a pure noise state $Y_{n,0} \sim \pi_d$, the noise is gradually transformed to approximate the data distribution $q_{n,\delta}$ for each worker-$n$. And $(Y_{n,t})_{t \in [0,T]}$ can be defined by the following SDE for each interval $t \in [t_k, t_{k+1}]$:

$$dY_{n,t} = \{Y_{n,t} + 2s_{\theta_R}(Y_{n,t_k}, T - t_k)\}dt + \sqrt{2}d\widetilde{B}_{n,t} \tag{10}$$

where $(\widetilde{B}_{n,t})_{t \in [0,T]}$ is a standard Brownian motion, and the marginals of this process can be denoted by $p_{n,t}$s. For $k = 0, \cdots, K-1$, the above (10) can be solved explicitly by

$$\begin{cases} Y_{n,t_{k+1}} = e^{\gamma_k} Y_{n,t_k} + 2(e^{\gamma_k} - 1)s_{\theta_R}(Y_{n,t_k}, T - t_k) \\ \qquad\qquad + \sqrt{e^{2\gamma_k} - 1} \cdot \epsilon_{n,k} \\ Y_{n,0} \sim \pi_d \end{cases}$$

where $\epsilon_{n,k} \sim \mathcal{N}(\mathbf{0}, \mathbf{I}_d)$. For additional details, please refer to Appendix B. An illustration of distributed diffusion model training with pruning is provided in Figure 1.

To provide a theoretical foundation for deploying diffusion models on geographically distributed and resource-constrained devices, we analyze the effects of incorporating pruning techniques into distributed diffusion model training in Section 4.

## 4. Main Results

In this section, we quantify the difference between the idealized data distribution $q_{n,\delta}$ generated by process (3) and the actual data distribution $p_{n,t_K}$ obtained from process (10), using KL divergence as the measurement metric.

Before presenting the formal results, we introduce the assumptions required for our theoretical analysis.

**Assumption 4.1** (Lipschitzian Gradient). Loss function $F_n(\cdot)$s are with Lipschitzian gradients. i.e., For $\forall \theta, \phi \in \mathbb{R}^D$, it holds that

$$\| \nabla F_n(\theta) - \nabla F_n(\phi) \| \le L \| \theta - \phi \| .$$

Assumption 4.1 (Lian et al., 2017) is commonly adopted to guarantee the stability and solvability of optimization problems. It ensures that the gradients of the loss functions vary smoothly, avoiding abrupt fluctuations, which is crucial for the convergence of gradient-based optimization algorithms.

**Assumption 4.2** (Pruning-induced Error). For an arbitrary mask $m_{n,r} \in \{0,1\}^D$ and a model $\theta_r \in \mathbb{R}^D$ ($r = 1, \cdots, R$ and $n = 1, \cdots, N$), we assume that there exists $w^2 \in [0,1)$:

$$\| \theta_r - \theta_r \odot m_{r,n} \|^2 \le w^2.$$

Assumption 4.2 (Qiao et al., 2023) ensures that the pruning operation, which typically involves using a binary mask, does not degrade the model's performance beyond a certain threshold.

**Assumption 4.3** (Bounded Variance). For any model $\theta$ and sample $\xi$, there exist $\sigma_1 > 0$ and $\sigma_2 > 0$:

$$\mathbb{E} \| \nabla f_n(\theta, \xi) - \nabla F_n(\theta) \|^2 \le \sigma_1^2,$$

$$\mathbb{E} \| \nabla F_n(\theta) - \nabla F(\theta) \|^2 \le \sigma_2^2.$$

Assumption 4.3 (Lian et al., 2017) imposes constraints on the influence of randomness, such as stochastic gradients, and regulates the divergence between local and global gradients. This ensures that these variations remain controlled and do not substantially disrupt the optimization process.

**Assumption 4.4** (Data Distribution). The data distribution $q_{n,0}$ of each worker-$n$ has finite second moments $M_{n,2}$.

Assumption 4.4 ensures that the data distribution of each worker has finite second moments, which is necessary for the convergence of the forward process in the diffusion model (Benton et al., 2024; Chen et al., 2023b;a).

Based on the above assumptions, our main results are established through the following steps:

**Step 1:** In Lemma 4.5, we establish an error bound for the discretization of the reverse SDE (3). This result extends the findings from previous work (Section 3.1 of (Benton et al., 2024)), which originally analyzed diffusion model training in a single-worker framework.

**Step 2:** We evaluate the impact of distributed collaboration on diffusion model training through a three-step analysis. First, we characterize the convergence behavior of distributed score function estimation in Lemma 4.6, capturing the effects of iterative updates, pruning operations, and stochastic errors. Next, leveraging this convergence behavior and the construction of auxiliary functions, we derive an upper bound on the local loss for each worker, as presented in Lemma 4.7. Finally, we examine the discrepancy between the ideal loss, given by

$$\sum_{k=0}^{K-1} \gamma_k \mathbb{E} \| s_{\theta_R}(Y_{n,t_k}, T - t_k) - \nabla \log q_{n,T-t_k}(Y_{n,t_k}) \|^2,$$

and the practical loss $F_n(\theta_R)$, which arises due to the application of the denoising score matching technique. This inconsistency is formally analyzed in Lemma 4.8.

**Step 3:** In Theorem 4.11, we quantify the discrepancy between the idealized data distribution $q_{n,\delta}$ and the actual data distribution $p_{n,t_K}$ using KL divergence. This discrepancy can be decomposed into two key components: the difference between reverse path measures and the impact of early stopping. The first component is derived by Steps 1 and 2 using Girsanov's theorem, as detailed in Lemma 4.9. The

second component captures the distance between $q_{n,T}$ and $\pi_d$ following the findings of previous studies (Chen et al., 2023a;b).

For the time discretization error, we provide the expected difference between the drift terms at different time.

**Lemma 4.5** (Time Discretization Error). *(Benton et al., 2024) Suppose $(Y_{n,t})_{t\in[0,T]}$ is the solution to the SDE (3), and there exists some $\kappa > 0$ such that for each discretized time point $t_0, \cdots, t_K$, we have $\gamma_k = t_{k+1} - t_k \leq \kappa \min\{1, T - t_{k+1}\}$. Then it holds that*

$$\sum_{k=0}^{K-1} \int_{t_k}^{t_{k+1}} \mathbb{E}[\| A_1 - A_2 \|^2] dt \lesssim \kappa^2 dK + \kappa M_{n,2} + \kappa dT$$

*where we write "$x \lesssim y$" to mean $x \leq Cy$ for an absolute constant $C > 0$. Additionally, the terms $A_1$ and $A_2$ are defined as $A_1 = \nabla \log q_{n,T-t}(Y_{n,t})$ and $A_2 = \nabla \log q_{n,T-t_k}(Y_{n,t_k})$, respectively.*

This proof, originally presented in Section 3.1 of (Benton et al., 2024), was developed in the context of single-worker diffusion model training. The core approach relies on a novel $It\hat{o}$ calculus argument, which establishes a differential inequality for $\mathbb{E}[\nabla \log q_{n,T-t}(Y_{n,t}) - \nabla \log q_{n,T-s}(Y_{n,s})]$.

Based on Assumptions 4.1-4.3, we can derive the following result for the distributed training of the score function estimation.

**Lemma 4.6** (Distributed Learning Dynamic). *If the local learning rate $\eta$ satisfies $0 < \eta \leq \min\{\sqrt{\frac{\Gamma^*}{640S^2L^2N}}, 1\}$, the following convergence result holds for the distributed learning of the above model $\theta_R$:*

$$\frac{1}{R} \sum_{r=0}^{R-1} \mathbb{E} \| \nabla F(\theta_r) \|^2 \leq \frac{8(F(\theta_0) - F(\theta_R))}{\eta SR} + (\sigma_1^2 + \sigma_2^2)$$
$$+ \frac{160w^2 LN}{\Gamma^*} + \frac{40N\sigma_2^2}{\Gamma^*} + \frac{16\eta LN\sigma_1^2}{(\Gamma^*)^2}$$

*where $F(\theta) = \frac{1}{N} \sum_{n=1}^{N} F_n(\theta)$ and $F_n(\theta)$ is defined in (5).*

*Proof Sketch.* We provide a brief outline here, and the detailed proof can be found in Appendix C.

Utilizing the Lipschitzian gradient assumption, we start the proof by analyzing the change in the global loss function during one round as the model transitions from $\theta_r$ to $\theta_{r+1}$:

$$\mathbb{E}[F(\theta_{r+1})] - \mathbb{E}[F(\theta_r)]$$
$$\leq \underbrace{\mathbb{E}\langle\nabla F(\theta_r), \theta_{r+1} - \theta_r\rangle}_{B_1^{(r)}} + \underbrace{\frac{L}{2}\mathbb{E} \| \theta_{r+1} - \theta_r \|^2}_{B_2^{(r)}} \quad (11)$$

Based on the local update (7) and the global model aggregation (8), $B_1^{(r)}$ and $B_2^{(r)}$ can be bounded as follows:

$$B_1^{(r)} \leq \frac{\eta L^2}{\Gamma^*} \sum_{n=1}^{N} \sum_{s=1}^{S} \mathbb{E} \| \theta_{r,n,s-1} - \theta_r \|^2 + \frac{\eta SN\sigma_2^2}{\Gamma^*}$$
$$- \frac{\eta S}{2}\mathbb{E} \| \nabla F(\theta_r) \|^2$$

$$B_2^{(r)} \leq \frac{2\eta^2 SLN\sigma_1^2}{(\Gamma^*)^2} + \frac{2\eta^2 SL^3}{\Gamma^*} \sum_{n=1}^{N} \sum_{s=1}^{S} \mathbb{E} \| \theta_{r,n,s-1} - \theta_r \|^2$$
$$+ \frac{2\eta^2 S^2 LN\sigma_2^2}{\Gamma^*} + 2\eta^2 S^2 L\mathbb{E} \| \nabla F(\theta_r) \|^2$$

To further derive $\mathbb{E} \| \theta_{r,n,s-1} - \theta_r \|^2$, we need to explore the cumulative entanglement of arbitrary pruning operations and local multistep training. In other words, it holds that

$$\mathbb{E} \| \theta_{r,n,s-1} - \theta_r \|^2 = \mathbb{E} \| \theta_{r,n,s-1} - \theta_{r,n,0} + \theta_{r,n,0} - \theta_r \|^2$$
$$\leq \underbrace{2\mathbb{E} \| \theta_{r,n,s-1} - \theta_{r,n,0} \|^2}_{B_3^{(r)}} + 2w^2 \quad (12)$$

where $B_3^{(r)}$ can be bounded as

$$B_3^{(r)} = 2\mathbb{E} \| -\eta \sum_{j=1}^{s-1} \nabla f_n(\theta_{r,n,j-1}, \xi_{n,j-1}) \odot m_{r,n} \|^2$$
$$\leq 2\eta^2(s-1) \sum_{j=1}^{s-1} \mathbb{E} \| \nabla f_n(\theta_{r,n,j-1}, \xi_{n,j-1}) -$$
$$\nabla F_n(\theta_{r,n,j-1}) + \nabla F_n(\theta_{r,n,j-1}) - F_n(\theta_r) +$$
$$\nabla F_n(\theta_r) - \nabla F(\theta_r) + \nabla F(\theta_r) \|^2$$
$$\leq 8\eta^2(s-1)L^2 \sum_{j=1}^{s-1} \mathbb{E} \| \theta_{r,n,j-1} - \theta_r \|^2 +$$
$$8\eta^2(s-1)^2\mathbb{E} \| F(\theta_r) \|^2 + 8\eta^2(s-1)^2(\sigma_1^2 + \sigma_2^2)$$

By summing (12) from $s = 1$ to $S$, from $n = 1$ to $N$, and from $r = 1$ to $R$, we can obtain the following inequality:

$$(1 - 8\eta^2 S^2 L^2) \sum_{r=0}^{R-1} \frac{1}{\Gamma^*} \sum_{n=1}^{N} \sum_{s=1}^{S} \mathbb{E} \| \theta_{r,n,s-1} - \theta_r \|^2$$
$$\leq \frac{8\eta^2 S^3 NR}{\Gamma^*}(\sigma_1^2 + \sigma_2^2) + \frac{8\eta^2 S^3 N}{\Gamma^*} \sum_{r=0}^{R-1} \mathbb{E} \| F(\theta_r) \|^2 +$$
$$\frac{2w^2 RSN}{\Gamma^*} \quad (13)$$

Then summing (12) from $r = 1$ to $R$, and substituting (13) to it, we can further control the appropriate learning rate $\eta$ to obtain the final result for distributed learning dynamic. This completes the proof outline. $\square$

Lemma 4.6 describes the rate at which the average gradient norm converges over all training rounds. The term $\frac{8(F(\theta_0)-F(\theta_R))}{\eta SR}$ reflects the impact of iterative updates on the convergence behavior, while the remaining terms capture the combined effects of pruning operations, randomness, and local errors.

Based on Lemma 4.6, we can further derive the following local loss bound:

**Lemma 4.7** (Local Loss Bound). *If the local learning rate $\eta$ satisfis $0 < \eta \le \min\{\sqrt{\frac{\Gamma^*}{640S^2L^2N}}, 1\}$, each local loss $F_n(\theta_R)$ can be bounded as*

$$F_n(\theta_R) \le \| F_n(\theta_0) - F(\theta_0) \| + \sigma_2 \| \theta_R - \theta_0 \| + F(\theta_0)$$
$$+ \frac{20\eta SRw^2 LN}{\Gamma^*} + \frac{5\eta SRN\sigma_2^2}{\Gamma^*} + \frac{2\eta^2 SRLN\sigma_1^2}{(\Gamma^*)^2}$$
$$+ \frac{\eta SR(\sigma_1^2 + \sigma_2^2)}{8}$$

*Proof.* Based on Lemma 4.6, we can directly obtain the following inequality based on the fact that the average gradient norm is non-negative:

$$F(\theta_R) \le F(\theta_0) + \frac{\eta SR(\sigma_1^2 + \sigma_2^2)}{8} + \frac{20\eta SRw^2 LN}{\Gamma^*} +$$
$$\frac{5\eta SRN\sigma_2^2}{\Gamma^*} + \frac{2\eta^2 SRLN\sigma_1^2}{(\Gamma^*)^2}$$

We now need to bound the discrepancy between local and global errors $\| F(\theta_R) - F_n(\theta_R) \|$. Consider the auxiliary function $h(t) = \theta_0 + t(\theta_R - \theta_0)$, then it holds that

$$F(\theta_R) - F(\theta_0) = \int_0^1 \nabla F(h(t))^T (\theta_R - \theta_0)dt$$

$$F_n(\theta_R) - F_n(\theta_0) = \int_0^1 \nabla F_n(h(t))^T (\theta_R - \theta_0)dt$$

Subtract the above two equations and take the norm to get

$$\| F_n(\theta_R) - F(\theta_R) \|$$
$$\le \| F_n(\theta_0) - F(\theta_0) \| + \sigma_2 \| \theta_R - \theta_0 \|$$

Then the proof can be completed by using the fact that $F_n(\theta_R) \le \| F_n(\theta_R) - F(\theta_R) \| + F(\theta_R)$. □

Lemma 4.7 describes the local loss bound of the score function estimation after $R$ rounds of collaboration. However, this result is based on the denoising score matching technique due to the unavailability of the score function $\nabla \log q_{n,T-t_k}(Y_{n,t_k})$. We explore its impact in Lemma 4.8:

**Lemma 4.8** (Equivalent Denoising Score Matching). *If $A_2 = \nabla \log q_{n,T-t_k}(Y_{n,t_k})$ as defined in Lemma 4.5, and $A_3 = s_{\theta_R}(Y_{n,t_k}, T - t_k)$, it holds that*

$$\sum_{k=0}^{K-1} \gamma_k \mathbb{E}[\| A_3 - A_2 \|^2] \le F_n(\theta_R) + C_1$$

*where $C_1$ is a constant independent of $\theta_R$.*

*Proof.*

$$\mathbb{E}[\| A_3 - A_2 \|^2]$$
$$= \mathbb{E} \| s_{\theta_R}(Y_{n,t_k}, T - t_k) \|^2 + \mathbb{E} \| \nabla \log q_{n,T-t_k}(Y_{n,t_k}) \|^2$$
$$- 2\mathbb{E}_{q_{n,0}}\mathbb{E}_{q_{n,T-t_k}|0}\langle s_{\theta_R}(Y_{n,t_k}, T - t_k), \nabla \log q(Y_{n,t_k}|X_{n,0})\rangle$$
$$= \mathbb{E} \| s_{\theta_R}(Y_{n,t_k}, T - t_k) \|^2 + \mathbb{E} \| \nabla \log q_{n,T-t_k}(Y_{n,t_k}) \|^2$$
$$+ 2\mathbb{E}\langle s_{\theta_R}(Y_{n,t_k}, T - t_k), \frac{Y_{n,t_k} - e^{-(T-t_k)}X_{n,0}}{1 - e^{-2(T-t_k)}}\rangle$$
$$= \mathbb{E} \| s_{\theta_R}(Y_{n,t_k}, T - t_k) + \frac{Y_{n,t_k} - e^{-(T-t_k)}X_{n,0}}{1 - e^{-2(T-t_k)}} \|^2$$
$$+ \mathbb{E} \| \nabla \log q_{n,T-t_k}(Y_{n,t_k}) \|^2 - \frac{d}{1 - e^{-2(T-t_k)}}$$
$$= \mathbb{E} \| s_{\theta_R}(Y_{n,t_k}, T - t_k) + \frac{Y_{n,t_k} - e^{-(T-t_k)}X_{n,0}}{1 - e^{-2(T-t_k)}} \|^2 + C_1^{t_k}$$
$$= \mathbb{E} \| s_{\theta_R}(Y_{n,t_k}, T - t_k) - \nabla \log q(Y_{n,t_k}|X_{n,0}) \|^2 + C_1^{t_k}$$

where we use the fact that $X_{n,t} = Y_{n,T-t}$ and $X_{n,t}|X_{n,0} \sim \mathcal{N}(X_{n,0}e^{-t}, (1 - e^{-2t})\boldsymbol{I}_d)$ in step 3. $C_1^{t_k}$ is a constant independent of $\theta_R$. Let $C_1 \ge \sum_{k=0}^{K-1} \gamma_k C_1^{t_k}$, it holds that

$$\sum_{k=0}^{K-1} \gamma_k \mathbb{E}[\| A_3 - A_2 \|^2]$$
$$\le \sum_{k=0}^{K-1} \gamma_k \mathbb{E} \| s_{\theta_R}(Y_{n,t_k}, T - t_k) - \nabla \log q(Y_{n,t_k}|X_{n,0}) \|^2$$
$$+ \sum_{k=0}^{K-1} \gamma_k C_1^{t_k} = F_n(\theta_R) + C_1$$

which completes the proof. □

Specially, as measures of learning loss, (4) and (6) are equivalent because the only difference between them is a constant.

Then we can bound the distance between path measures by Girsanov's theorem, which is the first part of Step 3.

**Lemma 4.9** (Distance between Path Measures). *Let $Q_n$, $P^{q_{n,T}}$ be the path measure of the solutions to the process (3) and (9). They both start from $Y_{n,0} \sim q_{n,T}$ and run from $t = 0$ to $t = t_K$. Then we establish the following result:*

$$\mathrm{KL}(Q_n \| P^{q_{n,T}}) \lesssim \sum_{k=0}^{K-1} \int_{t_k}^{t_{k+1}} \mathbb{E}[\| A_1 - A_2 \|^2]dt$$
$$+ \sum_{k=0}^{K-1} \gamma_k \mathbb{E}[\| A_3 - A_2 \|^2]$$

*where $A_1$, $A_2$, and $A_3$ are defined in Lemma 4.5 and Lemma 4.8.*

*Proof.* Based on Lemma 4.5 and 4.8, it can be directly derived that

$$\sum_{k=0}^{K-1} \int_{t_k}^{t_{k+1}} \mathbb{E}[\| A_1 - A_3 \|^2]\mathrm{d}t$$

$$\lesssim \sum_{k=0}^{K-1} \int_{t_k}^{t_{k+1}} \mathbb{E}[\| A_1 - A_2 \|^2]\mathrm{d}t + \sum_{k=0}^{K-1} \gamma_k \mathbb{E}[\| A_3 - A_2 \|^2]$$

$$< \infty$$

Then according to Girsanov's theorem, it holds that

$$\mathrm{KL}(Q_n \| P^{q_{n,T}}) \leq \sum_{k=0}^{K-1} \int_{t_k}^{t_{k+1}} \mathbb{E}[\| A_1 - A_3 \|^2]\mathrm{d}t.$$

This argument is essentially identical to Proposition 3 of the work (Benton et al., 2024) and Section 5.2 of the work (Chen et al., 2023b). □

The following Lemma 4.10 indicates that the generation error bound with early stopping can be decomposed into two parts: the distance between path measures and the distance between $q_{n,T}$ and $\pi_d$.

**Lemma 4.10** (Influence of Early Stopping). *Let $Q_n$, $P^{q_{n,T}}$ be the path measure of the solutions to the process (3) and (10) respectively. Under Assumption 4.4, the following result holds*

$$\mathrm{KL}(q_{n,\delta} \| p_{n,t_K}) \lesssim \mathrm{KL}(Q_n \| P^{q_{n,T}}) + (d + M_{n,2})e^{-2T}.$$

*Proof.* Since $q_{n,\delta}$ and $p_{n,t_K}$ are the pushfowards of the path measures $Q_n$ and $P^{q_{n,T}}$, it implies that $\mathrm{KL}(q_{n,\delta} \| p_{n,t_K}) \leq \mathrm{KL}(Q_n \| P^{\pi_d})$.

Based on Lemmas 4.9, 4.5 and 4.8, we see that $Q_n$ is absolutely continuous with respect to $P^{q_{n,T}}$ for each $n = 1, \cdots, N$. For a path $\boldsymbol{y} = (\boldsymbol{y}_t)_{t \in [0, t_K]}$, we can write $\frac{\mathrm{d}P^{q_{n,T}}}{\mathrm{d}P^{\pi_d}}(\boldsymbol{y}) = \frac{\mathrm{d}q_{n,T}}{\mathrm{d}\pi_d}(\boldsymbol{y}_0)$, which holds according to the fact $P^{q_{n,T}}$ and $P^{\pi_d}$ differ only by a change of starting distribution. Therefore,

$$\mathrm{KL}(Q_n \| P^{\pi_d}) = \mathbb{E}_{Q_n}\Big[ \log \big( \frac{\mathrm{d}P^{q_{n,T}}}{\mathrm{d}P^{\pi_d}}(Y) \frac{\mathrm{d}q_{n,T}}{\mathrm{d}\pi_d}(Y_0) \big) \Big]$$

$$= \mathrm{KL}(Q_n \| P^{q_{n,T}}) + \mathrm{KL}(P^{q_{n,T}} \| \pi_d)$$

where the first term can be bounded by Lemmas 4.9, 4.5 and 4.8, and for the second one, it can be controlled by the following inequality (Chen et al., 2023b):

$$\mathrm{KL}(P^{q_{n,T}} \| \pi_d) \lesssim (d + M_{n,2})e^{-2T}$$

which completes the proof. □

According to the above lemmas, the formal generation error bound can be summarized as follows.

**Theorem 4.11** (Generation Error Bound). *Suppose Assumptions 4.1-4.4 hold, $T \geq 1$, and there exists a constant $C_1$, and some $\kappa > 0$ such that $\gamma_k \leq \kappa \min\{1, T - t_{k+1}\}$. Then under the same settings of $\eta$ as in Lemma 4.7, for each worker-n, using the collaboratively learned model $\theta_R$, it yields the following result when approximating the data distribution $q_{n,\delta}$:*

$$\mathrm{KL}(q_{n,\delta} \| p_{n,t_K})$$
$$\lesssim \| F_n(\theta_0) - F(\theta_0) \| + \sigma_2 \| \theta_R - \theta_0 \| + F(\theta_0) + C_1 +$$
$$\frac{\eta SRw^2LN}{\Gamma^*} + \frac{\eta SRN(\sigma_1^2 + \sigma_2^2)}{\Gamma^*} + \frac{\eta^2 SRLN\sigma_1^2}{(\Gamma^*)^2} +$$
$$\kappa^2 dK + \kappa M_{n,2} + \kappa dT + (d + M_{n,2})e^{-2T}$$

In Theorem 4.11, the term $\| F_n(\theta_0) - F(\theta_0) \| + \sigma_2 \| \theta_R - \theta_0 \|$ captures the local-global error discrepancy. The term $C_1$ arises from denoising score matching, while $\kappa^2 dK + \kappa M_{n,2} + \kappa dT$ is due to time discretization approximations, and $(d + M_{n,2})e^{-2T}$ governs the convergence of the forward process. The remaining terms are interpreted as the local loss associated with $\theta_R$ which results from the distributed learning of score estimation with arbitrary pruning.

We next show that we can choose suitable hyperparameters to control the generation error bound based solely on the distributed training dynamics.

*Remark* 4.12 (Hyperparameter Selection). For $T \geq 1$, $\delta < 1$, $K \geq \log(1/\delta)$, if we set $\kappa = \Theta\left(\frac{T + \log(1/\delta)}{K}\right)$, $T = \frac{1}{2} \log\left(\frac{d + M_{n,2}}{F(\theta_0)}\right)$, $K = \Theta\left(\frac{(d + M_{n,2})(T + \log(1/\delta))^2}{F(\theta_0)}\right)$, and further control the learning rate to satisfy $\eta \leq \min\{\frac{F(\theta_0)\Gamma^*}{SRN(\sigma_1^2 + \sigma_2^2)}, \frac{F(\theta_0)\Gamma^*}{SRw^2L}, \sqrt{\frac{F(\theta_0)(\Gamma^*)^2}{SRNL\sigma_1^2}}\}$, we have $\mathrm{KL}(q_{n,\delta} \| p_{n,t_K}) \lesssim \| F_n(\theta_0) - F(\theta_0) \| + \sigma_2 \| \theta_R - \theta_0 \| + F(\theta_0) + C_1$.

Specially, if the initial samples of all workers are identically distributed, i.e., $q_{n,0} = q_{m,0}$ (for all $n, m = 1, \cdots, N$), then all local target loss function $F_n(\theta)$ are identical. In this context, each local loss $F_n(\theta_R)$ can be rewritten as

$$F_n(\theta_R) \lesssim F(\theta_0) + \frac{\eta SRw^2LN}{\Gamma^*} + \left(\frac{\eta}{\Gamma^*} + \frac{\eta^2 L}{(\Gamma^*)^2}\right)SRN\sigma_1^2$$

At a result, for the generation error bound, it holds that $\mathrm{KL}(q_{n,\delta} \| p_{n,t_K}) \lesssim F(\theta_0) + C_1$.

*Proof.* For $T \geq 1$, $\delta < 1$, $K \geq \log(1/\delta)$, if we set $\kappa = \Theta\left(\frac{T + \log(1/\delta)}{K}\right)$, then there obviously exists a sequence $\{t_k\}_{k=0}^K$ such that $\gamma_k \leq \kappa \min\{1, T - t_{k+1}\}$. Then, if we

set $K = \Theta\left(\frac{(d+M_{n,2})(T+\log(1/\delta))^2}{F(\theta_0)}\right)$, it holds

$$\begin{cases} \kappa^2 dK = \Theta\left(\dfrac{dF(\theta_0)}{d + M_{n,2}}\right) \lesssim F(\theta_0) \\[3mm] \kappa M_{n,2} = \Theta\left(\dfrac{M_{n,2}F(\theta_0)}{(d + M_{n,2})(T + \log(1/\delta))}\right) \lesssim F(\theta_0) \\[3mm] \kappa dT = \Theta\left(\dfrac{dTF(\theta_0)}{(d + M_{n,2})(T + \log(1/\delta))}\right) \lesssim F(\theta_0) \end{cases}$$

If we set $T = \frac{1}{2}\log\left(\frac{d+M_{n,2}}{F(\theta_0)}\right)$, it holds that $(d + M_{n,2})e^{-2T} = F(\theta_0)$. Then we have $\kappa^2 dK + \kappa M_{n,2} + \kappa dT + (d + M_{n,2})e^{-2T} \lesssim F(\theta_0)$.

Similarly, if we further control the learning rate to satisfy $\eta \leq \min\{\frac{F(\theta_0)\Gamma^*}{SRN(\sigma_1^2+\sigma_2^2)}, \frac{F(\theta_0)\Gamma^*}{SRNw^2L}, \sqrt{\frac{F(\theta_0)(\Gamma^*)^2}{SRNL\sigma_1^2}}\}$, we have $\frac{\eta SRw^2LN}{\Gamma^*} + \frac{\eta SRN(\sigma_1^2+\sigma_2^2)}{\Gamma^*} + \frac{\eta^2 SRLN\sigma_1^2}{(\Gamma^*)^2} \lesssim F(\theta_0)$.

These results complete the proof of Remark 4.12. □

## 5. Conclusion

In this paper, we introduced an efficient distributed diffusion model training mechanism that adapts to varying resource constraints via local sparse training. Theoretically, we established the first generation error bound for distributed diffusion models, which scales linearly with the data dimension $d$ and aligns with state-of-the-art results in the single-worker setting. Moreover, we demonstrated that with carefully selected hyperparameters, the generation performance of collaboratively trained diffusion models is primarily governed by the dynamics of distributed training.

## Acknowledgements

This work was supported in part by Joint Key Funds of National Natural Science Foundation of China under Grant U23A20302 and U24B20149, in part by the National Natural Science Foundation of China under Grant 62202273 and 62302247, in part by the National Key Research and Development Program of China under Grant No. 2022YFF0712100, in part by the Postdoctoral Fellowship Program of CPSF under Grant GZC20231460, in part by the China Postdoctoral Science Foundation under Grant 2024M761806.

## Impact Statement

This paper presents work whose goal is to advance the field of Machine Learning. There are many potential societal consequences of our work, none which we feel must be specifically highlighted here.

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

## A. Notation Table

In Table 1, we summarize the main notations in this paper.

*Table 1.* Some key notations and their descriptions.

| Notations | Descriptions |
|---|---|
| $T$ | The total time of noise scheduling |
| $t$ | The current time of noise scheduling |
| $K$ | The total number of discretized time interval of noise scheduling |
| $t_k$ | The $k$-th discretized time point of noise scheduling, and it holds $0 = t_0 < t_1 < t_2 < \cdots < t_K \leq T$ |
| $X_{n,t}$ | The data of worker-$n$ at time $t$ of noise scheduling, such as image data |
| $Y_{n,t}$ | The data of worker-$n$, which satisfies $Y_{n,t} = X_{n,T-t}$ |
| $q_{n,t}, t \in [0,T]$ | The marginals of the forward process (2) for each worker-$n$ |
| $d$ | The dimension of data |
| $(B_{n,t})_{t \in [0,T]}$ | The standard Brownian motion on $\mathbb{R}^d$ |
| $(\widetilde{B}_{n,t})_{t \in [0,T]}$ | The standard Brownian motion on $\mathbb{R}^d$ |
| $(B'_{n,t})_{t \in [0,T]}$ | The standard Brownian motion on $\mathbb{R}^d$ |
| $(\hat{B}_{n,t})_{t \in [0,T]}$ | The standard Brownian motion on $\mathbb{R}^d$ |
| $s_\theta(X_t, t)$ | The score approximation which can be parameterized by a neural network with a parameter vector $\theta \in \mathbb{R}^D$ |
| $D$ | The dimension of model parameter $\theta$, $\theta \in \mathbb{R}^D$ |
| $R$ | The total communication rounds for training the score approximation $s_\theta(\cdot)$ |
| $r$ | The current communication round for training the score approximation $s_\theta(\cdot)$ |
| $S$ | The number of local steps during two communication rounds |
| $N$ | The total number of workers |
| $N_r^{(i)}$ | The set of workers for which the value of coordinate-$i$ in the mask is non-zero, and $N_r^{(i)} = \{n : m_{r,n}^i = 1\}$ |
| $\Gamma^*$ | The minimum occurrences of any dimension parameter in the local model, and $\Gamma^* = \min_{r,i} |N_r^{(i)}| \geq 1$ |
| $f_n(\theta_{r,n,s}, \xi_{n,s})$ | The loss of worker-$n$ on a (a batch of) data sample $\xi_{n,s}$ in the step $s$ of round $r$ |
| $F_n(\theta)$ | The loss function of worker-$n$, and $F_n(\theta) = \mathbb{E}[f_n(\theta, \xi_n)]$ |
| $m_{r,n}$ | The local mask of worker-$n$ generated based on its own resources, and $m_{r,n} \in \{0,1\}^D$ |
| $\eta$ | The learning rate for training the score approximation $s_\theta(\cdot)$ |

## B. Solution to (10)

Consider (10):

$$dY_{n,t} = \{Y_{n,t} + 2s_{\theta_R}(Y_{n,t_k}, T - t_k)\}dt + \sqrt{2}d\widetilde{B}_{n,t}$$

We multiply both sides of (10) by $e^{-t}$ to get

$$d(e^{-t}Y_{n,t}) = -2s_{\theta_R}(Y_{n,t_k}, T - t_k)d(e^{-t}) + \sqrt{2}e^{-t}d\widetilde{B}_{n,t}$$

For each time interval $[t_k, t_{k+1}]$, we perform an integration operation to derive the following result:

$$e^{-t_{k+1}}Y_{n,t_{k+1}} = e^{-t_k}Y_{n,t_k} + 2s_{\theta_R}(Y_{n,t_k}, T - t_k)(e^{-t_k} - e^{-t_{k+1}}) + \sqrt{2}\int_{t_k}^{t_{k+1}} e^{-t}d\widetilde{B}_{n,t}$$

And then the following can be derived by multiplying both sides of the above equation by $e^{t_{k+1}}$:

$$Y_{n,t_{k+1}} = e^{\gamma_k}Y_{n,t_k} + 2(e^{\gamma_k} - 1)s_{\theta_R}(Y_{n,t_k}, T - t_k) + \sqrt{e^{2\gamma_k} - 1}\epsilon_{n,k}$$

where $\gamma_k = t_{k+1} - t_k$ and $\epsilon_{n,k} \sim \mathcal{N}(\mathbf{0}, \mathbf{I}_d)$.

## C. Detailed Proof of Lemma 4.6

Utilizing the Lipschitzian gradient assumption, we start the proof by analyzing the change in the loss function during one round as the model transitions from $\theta_r$ to $\theta_{r+1}$:

$$\mathbb{E}[F(\theta_{r+1})] - \mathbb{E}[F(\theta_r)] \leq \underbrace{\mathbb{E}\langle\nabla F(\theta_r), \theta_{r+1} - \theta_r\rangle}_{B_1^{(r)}} + \underbrace{\frac{L}{2}\mathbb{E}\parallel\theta_{r+1} - \theta_r\parallel^2}_{B_2^{(r)}} \tag{14}$$

Based on the local update (7) and the global model aggregation (8), it holds that

$$\theta_{r+1}^{(i)} - \theta_r^{(i)} = \frac{1}{|N_r^{(i)}|}\sum_{n\in N_r^{(i)}}\theta_{r,n,S}^{(i)} - \theta_r^{(i)}$$

$$= \frac{1}{|N_r^{(i)}|}\sum_{n\in N_r^{(i)}}\left(\theta_{r,n,0}^{(i)} - \eta\sum_{s=1}^S\nabla f_n^{(i)}(\theta_{r,n,s-1}, \xi_{n,s-1})\cdot m_{r,n}^{(i)}\right) - \theta_r^{(i)}$$

$$= \frac{1}{|N_r^{(i)}|}\sum_{n\in N_r^{(i)}}\left(\theta_r^{(i)}\cdot m_{r,n}^{(i)} - \eta\sum_{s=1}^S\nabla f_n^{(i)}(\theta_{r,n,s-1}, \xi_{n,s-1})\cdot m_{r,n}^{(i)}\right) - \theta_r^{(i)}$$

$$= -\eta\cdot\frac{1}{|N_r^{(i)}|}\sum_{n\in N_r^{(i)}}\sum_{s=1}^S\nabla f_n^{(i)}(\theta_{r,n,s-1}, \xi_{n,s-1})$$

where the last step follows from the fact that for all $n \in N_r^{(i)}$, $m_{r,n}^{(i)} = 1$.

Then $B_1^{(r)}$ and $B_2^{(r)}$ can be bounded as

$$B_1^{(r)} = \sum_{i=1}^D\mathbb{E}\langle\nabla F^{(i)}(\theta_r), \theta_{r+1}^{(i)} - \theta_r^{(i)}\rangle$$

$$= \sum_{i=1}^D\mathbb{E}\langle\nabla F^{(i)}(\theta_r), -\eta\cdot\frac{1}{|N_r^{(i)}|}\sum_{n\in N_r^{(i)}}\sum_{s=1}^S\nabla f_n^{(i)}(\theta_{r,n,s-1}, \xi_{n,s-1})\rangle$$

$$= \sum_{i=1}^D\mathbb{E}\langle\nabla F^{(i)}(\theta_r), -\eta\cdot\frac{1}{|N_r^{(i)}|}\sum_{n\in N_r^{(i)}}\sum_{s=1}^S\left(\nabla F_n^{(i)}(\theta_{r,n,s-1}) - \nabla F^{(i)}(\theta_r)\right)\rangle - \eta S\mathbb{E}\parallel\nabla F(\theta_r)\parallel^2$$

$$= -\eta S\mathbb{E}\parallel\nabla F(\theta_r)\parallel^2 - \eta S\sum_{i=1}^D\mathbb{E}\langle\nabla F^{(i)}(\theta_r), \frac{1}{S|N_r^{(i)}|}\sum_{n\in N_r^{(i)}}\sum_{s=1}^S\left(\nabla F_n^{(i)}(\theta_{r,n,s-1}) - \nabla F^{(i)}(\theta_r)\right)\rangle$$

$$\leq -\eta S\mathbb{E}\parallel\nabla F(\theta_r)\parallel^2 + \frac{\eta S}{2}\mathbb{E}\parallel\nabla F(\theta_r)\parallel^2$$

$$+ \frac{\eta S}{2}\sum_{i=1}^D\mathbb{E}\parallel\frac{1}{S|N_r^{(i)}|}\sum_{n\in N_r^{(i)}}\sum_{s=1}^S\left(\nabla F_n^{(i)}(\theta_{r,n,s-1}) - \nabla F_n^{(i)}(\theta_r) + \nabla F_n^{(i)}(\theta_r) - \nabla F^{(i)}(\theta_r)\right)\parallel^2$$

$$\leq -\frac{\eta S}{2}\mathbb{E}\parallel\nabla F(\theta_r)\parallel^2 + \frac{\eta}{2}\sum_{i=1}^D\frac{1}{|N_r^{(i)}|}\sum_{n\in N_r^{(i)}}\sum_{s=1}^S\mathbb{E}\parallel\nabla F_n^{(i)}(\theta_{r,n,s-1}) - \nabla F_n^{(i)}(\theta_r) + \nabla F_n^{(i)}(\theta_r) - \nabla F^{(i)}(\theta_r)\parallel^2$$

$$\leq -\frac{\eta S}{2}\mathbb{E}\parallel\nabla F(\theta_r)\parallel^2 + \frac{\eta}{\Gamma^*}\sum_{n=1}^N\sum_{s=1}^S\mathbb{E}\parallel\nabla F_n(\theta_{r,n,s-1}) - \nabla F_n(\theta_r)\parallel^2 + \frac{\eta}{\Gamma^*}\sum_{n=1}^N\sum_{s=1}^S\mathbb{E}\parallel\nabla F_n(\theta_r) - \nabla F(\theta_r)\parallel^2$$

$$\leq \frac{\eta L^2}{\Gamma^*}\sum_{n=1}^N\sum_{s=1}^S\mathbb{E}\parallel\theta_{r,n,s-1} - \theta_r\parallel^2 + \frac{\eta S N\sigma_2^2}{\Gamma^*} - \frac{\eta S}{2}\mathbb{E}\parallel\nabla F(\theta_r)\parallel^2$$

$$B_2^{(r)} = \frac{L}{2} \sum_{i=1}^{D} \mathbb{E} \parallel \theta_{r+1}^{(i)} - \theta_r^{(i)} \parallel^2$$

$$= \frac{L}{2} \sum_{i=1}^{D} \mathbb{E} \parallel -\eta \cdot \frac{1}{|N_r^{(i)}|} \sum_{n \in N_r^{(i)}} \sum_{s=1}^{S} \nabla f_n^{(i)}(\theta_{r,n,s-1}, \xi_{n,s-1}) \parallel^2$$

$$= \frac{L}{2} \sum_{i=1}^{D} \mathbb{E} \parallel -\eta \cdot \frac{1}{|N_r^{(i)}|} \sum_{n \in N_r^{(i)}} \sum_{s=1}^{S} \left( \nabla f_n^{(i)}(\theta_{r,n,s-1}, \xi_{n,s-1}) - \nabla F_n^{(i)}(\theta_{r,n,s-1}) + \nabla F_n^{(i)}(\theta_{r,n,s-1}) - \nabla F_n^{(i)}(\theta_r) \right.$$

$$\left. + \nabla F_n^{(i)}(\theta_r) - \nabla F^{(i)}(\theta_r) + \nabla F^{(i)}(\theta_r) \right) \parallel^2$$

$$\leq 2L \sum_{i=1}^{D} \mathbb{E} \parallel \frac{\eta}{|N_r^{(i)}|} \sum_{n \in N_r^{(i)}} \sum_{s=1}^{S} \left( \nabla f_n^{(i)}(\theta_{r,n,s-1}, \xi_{n,s-1}) - \nabla F_n^{(i)}(\theta_{r,n,s-1}) \right) \parallel^2 +$$

$$2L \sum_{i=1}^{D} \mathbb{E} \parallel \frac{\eta}{|N_r^{(i)}|} \sum_{n \in N_r^{(i)}} \sum_{s=1}^{S} \left( \nabla F_n^{(i)}(\theta_{r,n,s-1}) - \nabla F_n^{(i)}(\theta_r) \right) \parallel^2 +$$

$$2L \sum_{i=1}^{D} \mathbb{E} \parallel \frac{\eta}{|N_r^{(i)}|} \sum_{n \in N_r^{(i)}} \sum_{s=1}^{S} \left( \nabla F_n^{(i)}(\theta_r) - \nabla F^{(i)}(\theta_r) \right) \parallel^2 + 2L \sum_{i=1}^{D} \mathbb{E} \parallel \frac{\eta}{|N_r^{(i)}|} \sum_{n \in N_r^{(i)}} \sum_{s=1}^{S} \nabla F^{(i)}(\theta_r) \parallel^2$$

$$\leq \frac{2\eta^2 L}{(\Gamma^*)^2} \sum_{n=1}^{N} \sum_{s=1}^{S} \mathbb{E} \parallel \nabla f_n(\theta_{r,n,s-1}, \xi_{n,s-1}) - \nabla F_n(\theta_{r,n,s-1}) \parallel^2 + \frac{2\eta^2 SL}{\Gamma^*} \sum_{n=1}^{N} \sum_{s=1}^{S} \mathbb{E} \parallel \nabla F_n(\theta_{r,n,s-1}) - \nabla F_n(\theta_r) \parallel^2 +$$

$$\frac{2\eta^2 SL}{\Gamma^*} \sum_{n=1}^{N} \sum_{s=1}^{S} \mathbb{E} \parallel \nabla F_n(\theta_r) - \nabla F(\theta_r) \parallel^2 + 2\eta^2 S^2 L \mathbb{E} \parallel \nabla F(\theta_r) \parallel^2$$

$$\leq \frac{2\eta^2 SLN\sigma_1^2}{(\Gamma^*)^2} + \frac{2\eta^2 SL^3}{\Gamma^*} \sum_{n=1}^{N} \sum_{s=1}^{S} \mathbb{E} \parallel \theta_{r,n,s-1} - \theta_r \parallel^2 + \frac{2\eta^2 S^2 LN\sigma_2^2}{\Gamma^*} + 2\eta^2 S^2 L \mathbb{E} \parallel \nabla F(\theta_r) \parallel^2$$

Substituting $B_1^{(r)}$ and $B_2^{(r)}$ to (14), we can obtain

$$\mathbb{E}[F(\theta_{r+1})] - \mathbb{E}[F(\theta_r)]$$

$$\leq -\left( \frac{\eta S}{2} - 2\eta^2 S^2 L \right) \mathbb{E} \parallel \nabla F(\theta_r) \parallel^2 + \left( \frac{\eta L^2}{\Gamma^*} + \frac{2\eta^2 SL^3}{\Gamma^*} \right) \sum_{n=1}^{N} \sum_{s=1}^{S} \mathbb{E} \parallel \theta_{r,n,s-1} - \theta_r \parallel^2 +$$

$$\left( \frac{\eta SN}{\Gamma^*} + \frac{2\eta^2 S^2 LN}{\Gamma^*} \right) \sigma_2^2 + \frac{2\eta^2 SLN\sigma_1^2}{(\Gamma^*)^2} \tag{15}$$

Next, to further derive $\mathbb{E} \parallel \theta_{r,n,s-1} - \theta_r \parallel^2$, we need to explore the cumulative entanglement of arbitrary pruning operations and local multistep training. In other words, it holds that

$$\mathbb{E} \parallel \theta_{r,n,s-1} - \theta_r \parallel^2 = \mathbb{E} \parallel \theta_{r,n,s-1} - \theta_{r,n,0} + \theta_{r,n,0} - \theta_r \parallel^2$$

$$= 2\mathbb{E} \parallel \theta_{r,n,s-1} - \theta_{r,n,0} \parallel^2 + 2\mathbb{E} \parallel \theta_r \odot m_{r,n} - \theta_r \parallel^2$$

$$\leq \underbrace{2\mathbb{E} \parallel \theta_{r,n,s-1} - \theta_{r,n,0} \parallel^2}_{B_3^{(r)}} + 2w^2 \tag{16}$$

where $B_3^{(r)}$ can be bounded as

$$B_3^{(r)} = 2\mathbb{E} \parallel -\eta \sum_{j=1}^{s-1} \nabla f_n(\theta_{r,n,j-1}, \xi_{n,j-1}) \odot m_{r,n} \parallel^2$$

$$\leq 2\eta^2(s-1) \sum_{j=0}^{s} \mathbb{E} \parallel \nabla f_n(\theta_{r,n,j}, \xi_{n,j}) - \nabla F_n(\theta_{r,n,j}) + \nabla F_n(\theta_{r,n,j}) - \nabla F_n(\theta_r) + \nabla F_n(\theta_r) - \nabla F(\theta_r) + \nabla F(\theta_r) \parallel^2$$

$$\leq 8\eta^2(s-1)\sum_{j=1}^{s-1}\mathbb{E}\parallel \nabla F_n(\theta_{r,n,j-1})-\nabla F_n(\theta_r)\parallel^2 +8\eta^2(s-1)^2\mathbb{E}\parallel F(\theta_r)\parallel^2 +8\eta^2(s-1)^2(\sigma_1^2+\sigma_2^2)$$

$$\leq 8\eta^2(s-1)L^2\sum_{j=1}^{s-1}\mathbb{E}\parallel \theta_{r,n,j-1}-\theta_r\parallel^2 +8\eta^2(s-1)^2\mathbb{E}\parallel F(\theta_r)\parallel^2 +8\eta^2(s-1)^2(\sigma_1^2+\sigma_2^2)$$

By summing (16) from $s=1$ to $S$, from $n=1$ to $N$, and from $r=1$ to $R$, we can obtain the following inequality if the learning rate satisfies $\eta < \sqrt{\frac{1}{8S^2L^2}}$:

$$(1-8\eta^2S^2L^2)\sum_{r=0}^{R-1}\frac{1}{\Gamma^*}\sum_{n=1}^{N}\sum_{s=1}^{S}\mathbb{E}\parallel \theta_{r,n,s-1}-\theta_r\parallel^2 \leq \frac{8\eta^2S^3NR}{\Gamma^*}(\sigma_1^2+\sigma_2^2)+\frac{8\eta^2S^3N}{\Gamma^*}\sum_{r=0}^{R-1}\mathbb{E}\parallel F(\theta_r)\parallel^2 +\frac{2w^2RSN}{\Gamma^*}$$
(17)

Summing (14) from $r=0$ to $R-1$, we can obtain

$$\mathbb{E}[F(\theta_R)]-\mathbb{E}[F(\theta_0)]$$
$$\leq -(\frac{\eta S}{2}-2\eta^2S^2L)\sum_{r=0}^{R-1}\mathbb{E}\parallel \nabla F(\theta_r)\parallel^2 +\frac{\eta L^2+2\eta^2SL^3}{\Gamma^*(1-8\eta^2S^2L^2)}\Big(8\eta^2S^3NR(\sigma_1^2+\sigma_2^2)+8\eta^2S^3N\sum_{r=0}^{R-1}\mathbb{E}\parallel F(\theta_r)\parallel^2$$
$$+2w^2RSN\Big)+(\frac{\eta SN}{\Gamma^*}+\frac{2\eta^2S^2LN}{\Gamma^*})R\sigma_2^2+\frac{2\eta^2SLNR\sigma_1^2}{(\Gamma^*)^2}$$

Specially, we can further set the learning rate

$$\eta \leq \frac{1}{8SL} \quad \Leftrightarrow \quad \frac{1}{1-8\eta^2S^2L^2}\leq 8 \quad \Leftrightarrow \quad \eta SL \leq \frac{1}{8}$$

to get

$$\begin{cases} 2\eta^2S^2L \leq \eta SL\cdot 2\eta S = \dfrac{\eta S}{4} \\[2mm] 2\eta^2SL^3 \leq \eta SL\cdot 2\eta L^2 = \dfrac{\eta L^2}{4} \\[2mm] 2\eta^2S^2LN \leq \eta SL\cdot 2\eta SN = \dfrac{\eta SN}{4} \end{cases} \Rightarrow \quad \frac{\eta L^2+2\eta^2SL^3}{1-8\eta^2S^2L^2}\leq 8(\eta L^2+\frac{\eta L^2}{4})=10\eta L^2$$

Then

$$\mathbb{E}[F(\theta_R)]-\mathbb{E}[F(\theta_0)]$$
$$\leq -\frac{\eta S}{4}\sum_{r=0}^{R-1}\mathbb{E}\parallel \nabla F(\theta_r)\parallel^2 +\frac{10\eta L^2}{\Gamma^*}\Big(8\eta^2S^3NR(\sigma_1^2+\sigma_2^2)+8\eta^2S^3N\sum_{r=0}^{R-1}\mathbb{E}\parallel F(\theta_r)\parallel^2 +2w^2RSN\Big)+\frac{5\eta SNR\sigma_2^2}{4\Gamma^*}$$
$$+\frac{2\eta^2SLNR\sigma_1^2}{(\Gamma^*)^2}$$
$$= -(\frac{\eta S}{4}-\frac{80\eta^3S^3L^2N}{\Gamma^*})\sum_{r=0}^{R-1}\mathbb{E}\parallel \nabla F(\theta_r)\parallel^2 +\frac{80\eta^3S^3L^2NR(\sigma_1^2+\sigma_2^2)}{\Gamma^*}+\frac{20\eta L^2Sw^2NR}{\Gamma^*}+\frac{5\eta SNR\sigma_2^2}{4\Gamma^*}+\frac{2\eta^2SLNR\sigma_1^2}{(\Gamma^*)^2}$$
(18)

Let

$$\frac{\eta S}{4}-\frac{80\eta^3S^3L^2N}{\Gamma^*}\geq \frac{\eta S}{8} \quad \Leftrightarrow \quad \frac{80\eta^2S^2L^2N}{\Gamma^*}\leq \frac{1}{8} \quad \Leftrightarrow \quad \eta^2 \leq \frac{\Gamma^*}{640S^2L^2N}$$

It holds that

$$\frac{1}{R}\sum_{r=0}^{R-1}\mathbb{E}\parallel \nabla F(\theta_r)\parallel^2 \leq \frac{8(F(\theta_0)-F(\theta_R))}{\eta SR}+(\sigma_1^2+\sigma_2^2)+\frac{160w^2LN}{\Gamma^*}+\frac{40N\sigma_2^2}{\Gamma^*}+\frac{16\eta LN\sigma_1^2}{(\Gamma^*)^2}$$

which completes the proof.

*Figure 2.* Training loss of FedDM under the random pruning with different pruning levels.

## D. Experimental Details

### D.1. Experimental Setup

We conduct experiments using the Cifar-10 (Krizhevsky et al., 2009) SVHN (Netzer et al., 2011), and Fashion-MNIST (Xiao et al., 2017) datasets. To simulate a distributed learning scenario, we partition the training data among 10 workers. As described in Section 3, DDPM (Ho et al., 2020) can be viewed as a special case of our work, so we consider its distributed version (known as FedDM (Vora et al., 2024)) under resource-constrained conditions. In the experiments, we mainly consider two pruning techniques: Random Pruning (R) and Top-k Pruning (T) based on model weight. In particular, in order to explore the heterogeneity of pruning policy caused by resource differences among workers, we set for different pruning levels named F (Full), L (Large), M (Medium) and S (Small):

- **F:** All workers with full model;

- **L:** $80\%$ workers with full model, and $20\%$ workers with $75\%$ model parameters;

- **M:** $60\%$ workers with full model, $20\%$ workers with $80\%$ model parameters, and $20\%$ workers with $75\%$ model parameters;

- **S:** $60\%$ workers with full model, and $40\%$ workers with $75\%$ model parameters.

We utilize multiple metrics to evaluate the performance of distributed training diffusion models with different pruning levels: Training loss is used to assess the convergence for distributed learning of score estimation. Additionally, the Inception Score (IS) and Fréchet Inception Distance (FID) are employed to evaluate the quality of data generation.

In the training stage of obtaining a score estimation, we use the U-Net backbone containing residual blocks (Tun et al., 2023). And we use the following settings unless otherwise stated: The number of communication rounds $Q$ is set as 300, the local training steps $S$ are configured as 5 epochs for Cifar-10 and 2 epochs for both SVHN and Fashion-MNIST, and the step size $\eta$ is 0.0001.

All the experiments are implemented in PyTorch 2.5.1, Python 3.12, Cuda 12.1. And we run them on a Cloud Server with Intel(R) Xeon(R) Platinum 8358P CPU and total 10 RTX 3090 GPUs in Ubuntu 22.04.

### D.2. Model convergence for distributed learning of score estimation

We assess the convergence for distributed learning of score estimation on the above three datasets, using Random (R) and Top-k (T) pruning techniques. Specifically, we establish four pruning levels (F, L, M, and S) to observe the effects on convergence behavior. This series of experiments is designed to systematically evaluate how various levels of model sparsity influence the training dynamics.

Figures 2 and 3 illustrate the impact of different pruning strategies and pruning levels on the convergence rate of the distributed training diffusion model across three datasets. Overall, the training loss in all settings is effectively reduced as the number of communication rounds increases, verifying the effectiveness of the coordinate-wise aggregation method. Under both pruning strategies, as the degree of pruning increases (denoted by F, L, M, S), the training loss requires more

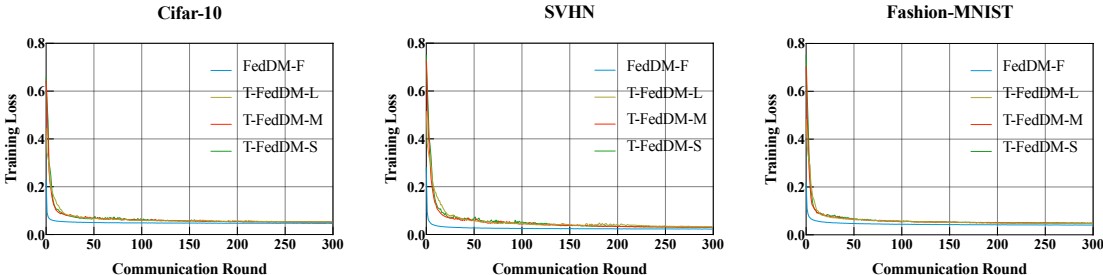

*Figure 3.* Training loss of FedDM under the Top-k pruning with different pruning levels.

*Table 2.* IS and FID comparison of FedDM with different pruning levels.

| Method | Cifar-10 | | SVHN | | Fashion-MNIST | |
|---|---|---|---|---|---|---|
| | IS ($\uparrow$) | FID ($\downarrow$) | IS ($\uparrow$) | FID ($\downarrow$) | IS ($\uparrow$) | FID ($\downarrow$) |
| FedDM-F | $4.59 \pm 0.13$ | 73.73 | $2.79 \pm 0.04$ | 163.36 | $3.58 \pm 0.08$ | 87.59 |
| R-FedDM-L | $3.95 \pm 0.12$ | 103.59 | $2.76 \pm 0.04$ | 93.78 | $3.47 \pm 0.04$ | 53.70 |
| R-FedDM-M | $4.01 \pm 0.08$ | 104.53 | $2.60 \pm 0.04$ | 127.47 | $3.32 \pm 0.08$ | 52.31 |
| R-FedDM-S | $3.60 \pm 0.07$ | 111.21 | $2.53 \pm 0.05$ | 120.57 | $3.46 \pm 0.07$ | 49.94 |
| T-FedDM-L | $4.39 \pm 0.08$ | 83.75 | $2.72 \pm 0.04$ | 157.19 | $3.59 \pm 0.07$ | 87.85 |
| T-FedDM-M | $4.54 \pm 0.10$ | 80.42 | $2.55 \pm 0.05$ | 146.27 | $3.54 \pm 0.06$ | 100.69 |
| T-FedDM-S | $4.31 \pm 0.13$ | 84.98 | $2.51 \pm 0.06$ | 193.84 | $3.63 \pm 0.07$ | 109.83 |

communication rounds to decrease effectively, and the total reduction diminishes. This is because the reduced model introduces additional errors, which slows the convergence rate to a certain extent.

### D.3. Data Generation Quality

We assess the performance of distributed training DDPM (known as FedDM) with different pruning levels on the above three datasets. Specifically, we establish four pruning levels (F, L, M, and S) and utilize two indicators, IS and FID, to observe and compare the average data generation quality.

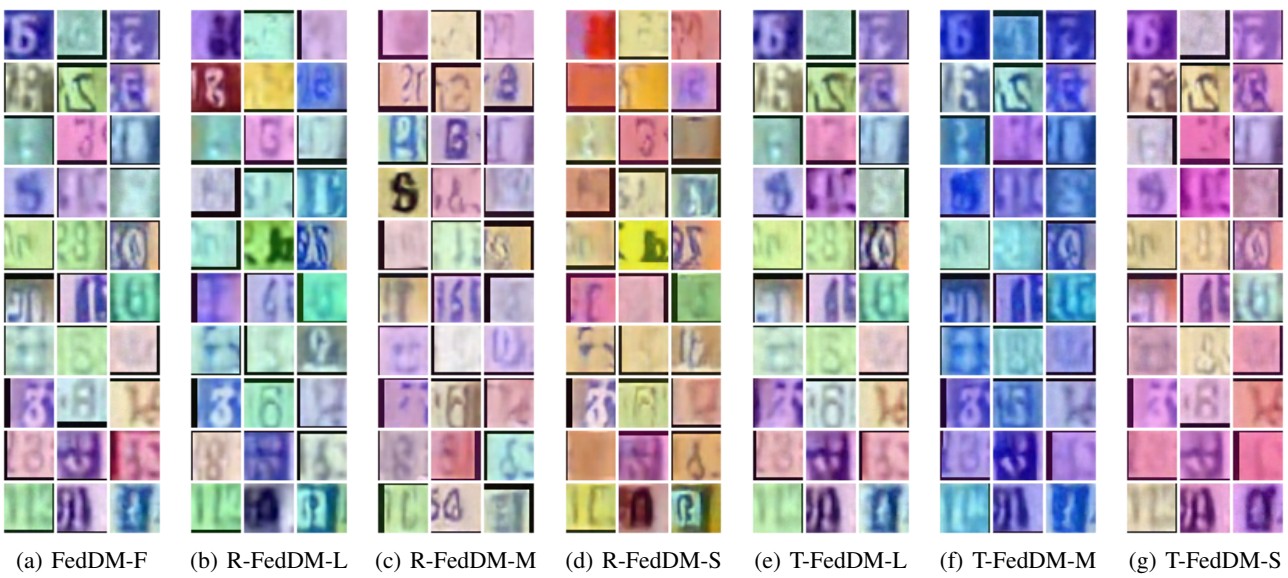

(a) FedDM-F  (b) R-FedDM-L  (c) R-FedDM-M  (d) R-FedDM-S  (e) T-FedDM-L  (f) T-FedDM-M  (g) T-FedDM-S

*Figure 5.* Generated samples on SVHN for each pruning setting of the two pruning techniques.

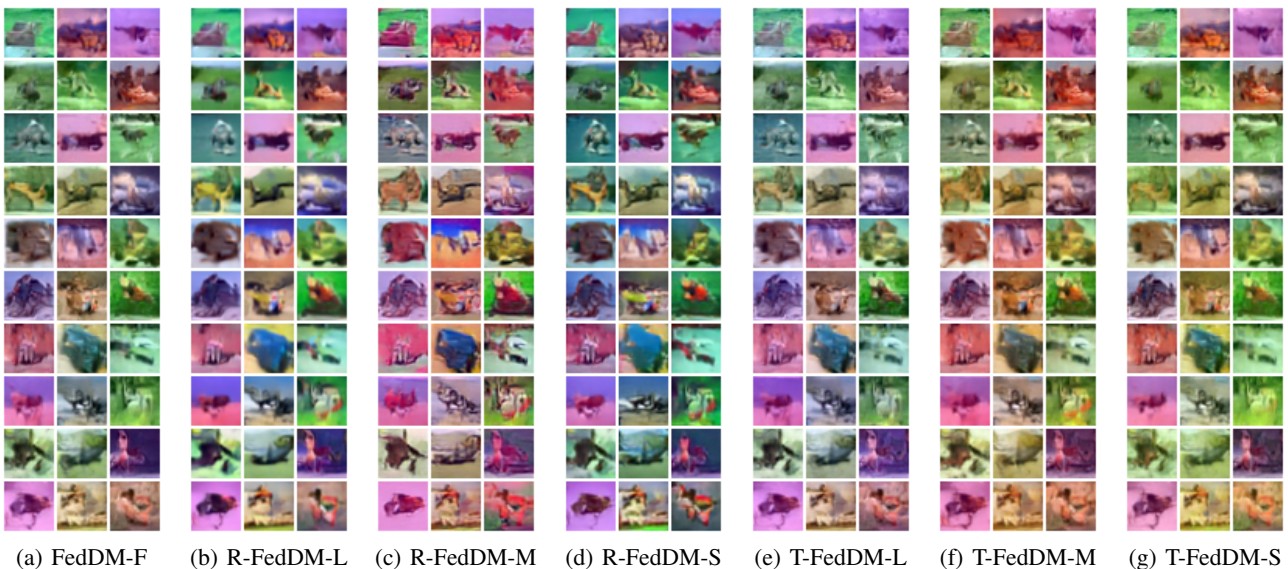

| (a) FedDM-F | (b) R-FedDM-L | (c) R-FedDM-M | (d) R-FedDM-S | (e) T-FedDM-L | (f) T-FedDM-M | (g) T-FedDM-S |

*Figure 4.* Generated samples on CIFAR-10 for each pruning setting of the two pruning techniques.

As shown in Table 2, the experimental results demonstrate that pruning significantly impacts the performance of diffusion models in distributed learning, with the effects closely related to the pruning strategy, dataset complexity, and model heterogeneity. On complex datasets such as CIFAR-10 and SVHN, the full model (FedDM-F) achieves the best performance, while increased pruning levels lead to a substantial decline in the quality of random pruning (R-FedDM), as indicated by decreased IS scores and increased FID values, particularly at high pruning levels (e.g., S). In contrast, Top-k pruning (T-FedDM) better preserves model performance by retaining critical parameters, resulting in smaller increases in FID and performance closer to the full model, especially at moderate pruning levels (e.g., M). For simpler datasets like Fashion-MNIST, where the data distribution is less complex, pruning has a relatively smaller impact, and the performance difference between random pruning and Top-k pruning is minimal. Additionally, on Fashion-MNIST, higher pruning levels unexpectedly improve FID values. This phenomenon can be attributed to the lower capacity requirements of simple data distributions, where high pruning reduces redundant parameters, acting as a regularization effect to prevent overfitting, thus smoothing the generated distribution and making it closer to the real distribution. Model heterogeneity introduced by pruning is another critical factor affecting global performance, with random pruning more likely to cause aggregation errors, while Top-k pruning alleviates this issue to some extent. Overall, Top-k pruning proves more advantageous for complex datasets, while random pruning is better suited for resource-constrained scenarios involving simpler tasks.

The generated samples shown in Figures 4-6 further reveal that pruning influences generation quality, particularly on complex datasets. For CIFAR-10 and SVHN, the full model produces images with more details and consistent color distribution, whereas aggressive pruning leads to noticeable degradations—such as blurred features, increased noise, and color distortions. Top-k pruning preserves critical parameters more effectively, yielding images that are more closed to those produced from the full model, particularly under moderate pruning conditions. For simpler datasets like Fashion-MNIST, the impact of pruning is less severe; in fact, higher pruning levels sometimes result in better images due to a regularization effect that eliminates redundant details and prevents overfitting. Future work can focus on optimizing pruning strategies and aggregation algorithms to further balance model efficiency and performance across various data distributions and task requirements.

## E. Discussion

Our analysis relies on several standard assumptions commonly adopted in the distributed learning literature, such as bounded gradient variance. While these assumptions facilitate tractable theoretical analysis, they may not fully capture the complexities of real-world distributed systems.

In our theoretical results, the term $w^2$ reflects the extent of pruning applied to the distributed model. As pruning becomes

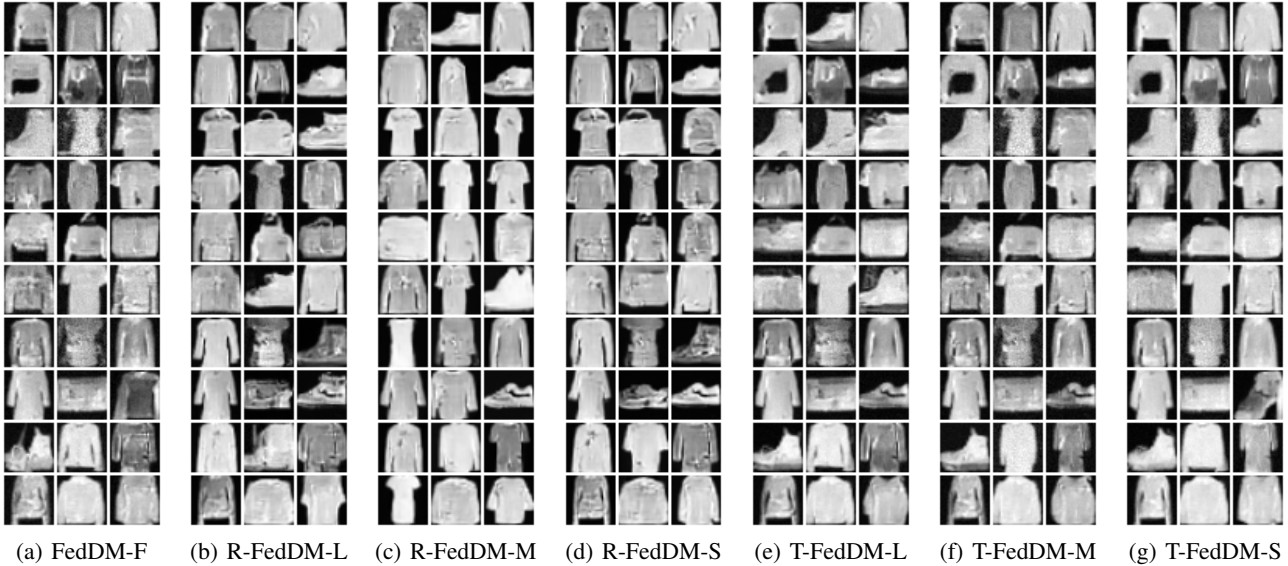

   (a) FedDM-F    (b) R-FedDM-L    (c) R-FedDM-M    (d) R-FedDM-S    (e) T-FedDM-L    (f) T-FedDM-M    (g) T-FedDM-S

*Figure 6.* Generated samples on Fashion-MNIST for each pruning setting of the two pruning techniques.

more aggressive (i.e., fewer parameters are retained), this term increases, leading to a looser error bound. The parameter $\Gamma^*$ captures the frequency with which each model parameter is updated across the distributed workers. A small $\Gamma^*$ indicates that some parameters are rarely trained, which may lead to suboptimal or imbalanced updates and thus a larger bound. This motivates us to seek more effective pruning strategies that achieve a balance between resource availability and generation quality, which we leave for the future.

While this work provides a theoretical perspective for distributed diffusion models under resource constraints, real-world constraints are often more complex. These constraints include heterogeneity in computational power, memory availability, communication latency, and the frequency of parameter updates. Such variability can pose significant challenges for maintaining model quality. Extending our framework to explicitly model these factors—such as by incorporating asynchronous optimization or adaptive, resource-aware pruning—also represents an important direction for future research.

