# OpenReview forum: "How Distributed Collaboration Influences the Diffusion Model Training? A Theoretical Perspective"
_ICML.cc/2025/Conference — ICML 2025 poster_

### Official Review · Reviewer_YCna · 2025-03-10

**Overall Recommendation:** 3

**Summary:**

The authors explore the theoretical performance of distributed diffusion models, particularly in environments where computational resources and data availability vary across workers. The authors establish a generation error bound for distributed diffusion models under resource constraints, demonstrating a linear relationship with the data dimension and alignment with existing single-worker results. Key contributions include a novel training mechanism that maintains data privacy via synchronized noise scheduling and sparse training strategies, an analysis of the impact of hyperparameter selection on model performance, and a theoretical framework for optimizing distributed diffusion models.

**Claims And Evidence:**

The claims in the paper are generally well-supported by theoretical derivations and empirical validation. The generation error bound is rigorously formulated using mathematical proofs, and its consistency with single-worker results enhances its credibility. The experimental evaluation demonstrates the practical viability of the proposed training method by assessing convergence behavior and data generation quality under different pruning strategies.

**Essential References Not Discussed:**

The paper cites most of the relevant literature but could consider discussing more recent advances in federated learning techniques for generative models, particularly those addressing resource efficiency.

**Experimental Designs Or Analyses:**

The experimental setup effectively simulates real-world distributed training scenarios, using datasets such as CIFAR-10, SVHN, and Fashion-MNIST. The study examines different pruning strategies (random and top-k pruning) and their impact on training loss and data generation quality. The inclusion of multiple pruning levels ensures a comprehensive analysis.

Although this is a theoretical paper, further exploration of how more diverse resource constraints affect the performance of distributed diffusion models will enhance the paper's persuasiveness.

**Methods And Evaluation Criteria:**

The methods used in the paper, including distributed diffusion models training with noise scheduling and sparse model updates, align well with the challenges posed by heterogeneous computing environments. The evaluation criteria are reasonable, including convergence analysis, Inception Score (IS), and Frechet Inception Distance (FID), which are commonly used for assessing generative models.

**Other Comments Or Suggestions:**

1. Consider providing a high-level summary of theoretical results for accessibility.
2. The author should further explain how the constraints on hyperparameters are derived.

**Other Strengths And Weaknesses:**

Strengths:
1. It establishes the first known generation error bound for distributed diffusion models under resource constraints, a significant theoretical contribution that advances our understanding of model performance in distributed settings.
2. The mathematical proofs are comprehensive and well-structured, utilizing advanced techniques such as Girsanov’s theorem to quantify training errors, making the study highly rigorous.
3. The paper employs advanced mathematical tools, such as Girsanov’s theorem, to derive a precise error bound. The assumptions made (Lipschitz continuity, bounded variance, pruning-induced errors) are reasonable and well-motivated. Theoretical claims are clearly stated and justified, making them highly credible.
4. The paper is well-structured, with clear mathematical notations and logical flow, making it easy to follow.

Weaknesses:
1. The results of the hyperparameter selection are not intuitive, and the authors could add more discussion to explain it in detail.
2. The authors do not explicitly discuss the limitations of their proposed approach or suggest potential future research directions. While the study provides strong theoretical and empirical contributions, a clear discussion on the scope and boundaries of the proposed method would help contextualize its impact and guide future work.

**Questions For Authors:**

1. The paper states that the derived error bound is consistent with single-worker results. Could the authors elaborate on the key similarities and differences in how errors accumulate in distributed vs. single-worker settings?
2. What real-world applications might this work provide theoretical guidance for? Adding a corresponding discussion would benefit the paper.

**Relation To Broader Scientific Literature:**

This work is well-positioned in the context of distributed machine learning and generative modeling. It builds upon foundational studies in diffusion models and distributed optimization, such as prior work on stochastic control and Girsanov-based error bounds.

**Theoretical Claims:**

The theoretical claims, particularly the derivation of the generation error bound, are well-supported. The proofs follow established methodologies, such as using Girsanov’s theorem for error quantification. The assumptions made (e.g., Lipschitz continuity, pruning-induced error constraints) are reasonable and align with prior work.

While the proofs are detailed, it would be beneficial to provide more intuition behind the key theoretical results for better accessibility.

---

> ### Author Rebuttal · Authors · 2025-03-31
>
> We thank the Reviewer YCna for the time and valuable feedback! We would try our best to address the comments one by one.
>
> **Response to “Theoretical Claims”:**  We agree that supplementing technical proofs with intuitive explanations greatly enhances accessibility. And we have added some explanation to the key results. For example, regarding the reverse SDE derivation, we now clarify that reversing the SDE entails “rewinding” the forward process by inverting the drift term, which is conceptually consistent with the notion of reconstructing the data distribution from noise.
>
> **Response to “Experimental Designs Or Analyses”:** We have added a discussion regarding the limitations and future directions in the appendix. Specifically, we offer the following reflections: While this work provides a theoretical perspective for distributed diffusion models under resource constraints, real-world constraints are often more complex. These constraints include heterogeneity in computational power, memory availability, communication latency, and the frequency of parameter updates. Such variability can pose significant challenges for maintaining model quality. Extending our framework to explicitly model these factors—such as by incorporating asynchronous optimization or adaptive, resource-aware pruning—represents an important direction for future research.
>
> **Response to “Other Weaknesses 1” ＆ “Other Comments Or Suggestions 2”:** We have added a detailed derivation for Remark 4.12 in the Appendix. For more details, you can refer to our response to Reviewer ySb9.
>
> **Response to “Other Weaknesses 2” ＆ “Other Comments Or Suggestions 1”:** We have added a discussion on the important theoretical results, limitations and future directions in the appendix. The details are as follows:
>
> Our analysis relies on several standard assumptions commonly adopted in the distributed learning literature, such as bounded gradient variance. While these assumptions facilitate tractable theoretical analysis, they may not fully capture the complexities of real-world distributed systems.
>
> In our theoretical results, the term $w^2$ reflects the extent of pruning applied to the distributed model. As pruning becomes more aggressive (i.e., fewer parameters are retained), this term increases, leading to a looser error bound. The parameter $\\Gamma ^*$ captures the frequency with which each model parameter is updated across the distributed workers. A small $\\Gamma^{*}$ indicates that some parameters are rarely trained, which may lead to suboptimal or imbalanced updates and thus a larger bound. This motivates us to seek more effective pruning strategies that achieve a balance between resource availability and generation quality, which we leave for the future.
>
> While this work provides a theoretical perspective for distributed diffusion models under resource constraints, real-world constraints are often more complex. These constraints include heterogeneity in computational power, memory availability, communication latency, and the frequency of parameter updates. Such variability can pose significant challenges for maintaining model quality. Extending our framework to explicitly model these factors—such as by incorporating asynchronous optimization or adaptive, resource-aware pruning—also represents an important direction for future research.
>
> **Response to “Questions For Authors 1”:** While our error bound shares a structural similarity with those in single-worker settings—relying on factors such as model dimension—there are key differences in how errors accumulate. Specially, distributed systems introduce additional challenges due to inter-worker heterogeneity and uneven parameter training. These are captured in our analysis by terms like $\\sigma _ 2^2$ and  $\\Gamma^*$, which do not appear in single-worker settings. Moreover, pruning may lead to imbalanced updates, which is quantified by the terms $w^2$ and $\\Gamma^*$ in our bound. In this context, the assumption of perfect score function estimation—often made in single-worker analyses—no longer holds. The key contribution of our theoretical framework is to explicitly characterize new errors that are unique to distributed generative modeling.
>
>
> **Response to “Questions For Authors 2”:** We agree that identifying real-world applications can help illustrate our theoretical contributions. However, to maintain the paper’s main theoretical focus, we refrain from expanding on specific applications within the main text. Nonetheless, we briefly highlight several potential application areas where our theoretical findings may provide guidance: (1) Federated generative modeling in privacy-sensitive domains such as healthcare and finance. (2) Collaborative generation across edge devices for generative tasks like image synthesis or speech enhancement, where each device has limited compute/memory.
>
> If there are any further confusions, we are happy to clarify them. Thank you again for your recognition of our work.

---

### Official Review · Reviewer_ySb9 · 2025-03-10

**Overall Recommendation:** 4

**Summary:**

This paper presents a theoretical analysis of distributed diffusion model training in scenarios where computational resources and data availability vary across workers. Traditional single-worker diffusion models assume uniform resources and centralized data, which are impractical in distributed settings. To address this, the authors introduce a novel distributed training mechanism that preserves data privacy through synchronized noise scheduling and accommodates resource heterogeneity via local sparse training. They establish the first generation error bound for distributed diffusion models, demonstrating a linear relationship with the data dimension d and consistency with state-of-the-art single-worker results. Using Girsanov’s theorem, the paper quantifies the impact of time discretization, training sparsity, and data heterogeneity on model generation quality. Additionally, the authors analyze coordinate-wise model aggregation to mitigate sparse training errors and show that hyperparameter selection, particularly learning rate and noise scheduling, plays a crucial role in optimizing distributed training dynamics. The findings provide a solid theoretical foundation for extending diffusion models to decentralized environments while maintaining competitive performance.

**Claims And Evidence:**

The claims are supported by rigorous proofs, and the key claims and their corresponding evidence are as follows:
Claim1 : Distributed training of diffusion models introduces unique challenges due to resource heterogeneity and data privacy concerns.
Evidence 1: The paper discusses how centralized data processing is impractical in real-world scenarios and introduces a distributed training mechanism that allows for local sparse training while preserving privacy. This is mathematically formalized in the distributed learning dynamics (Section 3), where the authors account for the effects of worker variability and introduce coordinate-wise model aggregation.
Claim 2: A generation error bound for distributed diffusion models is derived, which scales linearly with the data dimension d and aligns with single-worker results.
Evidence 2: The theoretical generation error bound is rigorously derived in Theorem 4.11, which decomposes the total error into components from time discretization, distributed training errors, and early stopping. The proof relies on Girsanov’s theorem and previous results from single-worker diffusion models, showing that the bound remains consistent with known results while extending to the distributed setting.
Claim 3: Hyperparameter selection, particularly noise scheduling and learning rate, significantly impacts the generation quality in distributed settings.
Evidence3: The paper provides a hyperparameter selection strategy (Remark 4.12) to control the generation error bound. By carefully tuning these parameters, the dominant factor influencing model quality remains the distributed training dynamics. This is supported by analytical expressions that show how different choices affect convergence.

**Essential References Not Discussed:**

It cites the most relevant theoretical works on diffusion model error bounds, distributed optimization, and federated learning, ensuring a strong foundation for its contributions. However, while this paper is primarily theoretical, incorporating discussions on experimental works related to distributed diffusion models would further highlight the importance of this theoretical analysis.

**Experimental Designs Or Analyses:**

As a theoretical paper, the primary contribution lies in the mathematical analysis of distributed diffusion model training, and the experimental results in the appendix serve as a supplementary aid to help understand the practical implications of distributed training under resource constraints. The experimental design focuses on how pruning strategies and pruning levels affect training loss and generation quality, providing empirical insights into the role of sparse training and coordinate-wise aggregation and effectively supporting the theoretical claims.

**Methods And Evaluation Criteria:**

The authors present a rigorous theoretical framework for analyzing distributed diffusion model training, leveraging coordinate-wise aggregation, sparse training, and hyperparameter optimization to address key challenges in distributed settings. The use of Girsanov’s theorem and score matching techniques ensures a solid mathematical foundation, and the derivation of the generation error bound provides valuable insights into the effects of time discretization, distributed training dynamics, and early stopping. While the theoretical analysis is comprehensive, the assumptions of bounded variance may not always hold in real-world settings, and discussing their relaxation would strengthen the generalizability of the results. Overall, the proposed methods and evaluation criteria are well-suited for a theoretical study, and the findings provide a solid foundation for future research on distributed generative models.

**Other Comments Or Suggestions:**

1. While the notation is mathematically rigorous, some equations, particularly in the reverse SDE derivation and loss function formulation, may benefit from a brief intuitive explanation to make the theoretical framework more accessible.

2. The paper highlights the role of learning rate and noise scheduling in distributed diffusion training, but a brief guideline or theoretical intuition on hyperparameter choices would be useful for researchers implementing these ideas.

**Other Strengths And Weaknesses:**

Pros.:

— The paper is the first to establish a generation error bound for distributed diffusion models. By demonstrating that the error bound scales linearly with data dimension d and remains consistent with existing single-worker results, the paper provides a foundational contribution to the theory of distributed generative modeling.

— The theoretical claims are well-structured, logically derived, and supported by rigorous proofs, making the results convincing and mathematically sound. The decomposition of the generation error into multiple contributing factors (time discretization, distributed training dynamics, early stopping, etc.) helps provide a deeper understanding of the challenges in distributed diffusion model training.

— While the paper is primarily theoretical, it addresses key challenges in real-world distributed learning, including data heterogeneity, resource variability, and privacy constraints.

Cons.:

— Some of the notation choices, particularly in the reverse SDE derivation and loss function formulation, might require prior familiarity with score-based generative modeling. While the notation table in the appendix helps, adding a brief intuitive explanation of key equations in the main text could improve readability.

— This paper emphasizes the importance of learning rate and noise scheduling, but only gives the results in the Remark. However, as far as I am concerned, the results of these hyperparameter selections are not intuitive, and adding more details will improve persuasiveness.

**Questions For Authors:**

1. In Remark 4.12, for the iid setting of data, does the bound of each local loss omit the constant C1, and shouldn't sigma2 be zero when all F_n are the same?

2. In Theorem 4.11, the generated error bound is affected by both the pruning level w^2 and the minimum parameter occurrences \Gamma. Can the authors explain this in more detail? And discuss the limitations of the current work?

3. The results highlight the role of learning rate and noise scheduling in distributed training. Are there any general guidelines for selecting these hyperparameters to ensure the error bound remains tight? In other words, how is the choice of hyperparameters established in Remark4.12?

**Relation To Broader Scientific Literature:**

The authors extend the theoretical study of diffusion models from single-worker settings to distributed training under resource constraints, contributing to both generative modeling and distributed optimization literature. Prior research has established error bounds, stability, and convergence properties for single-worker diffusion models (e.g., Benton et al., 2024; Chen et al., 2022, 2023), while this work introduces the first generation error bound for distributed diffusion models, demonstrating that the error maintains a linear dependency on data dimension d. By bridging the gap between diffusion model theory and distributed optimization, this paper provides a foundational step toward resource-aware generative models in distributed environments.

**Theoretical Claims:**

The theoretical claims in the paper are supported by rigorous mathematical derivations, particularly in establishing the generation error bound for distributed diffusion models. I examined the key proofs, including:
1. Theorem 4.11 is the core theoretical result, providing an upper bound on the difference between the ideal and actual data distributions in the KL divergence sense. The proof systematically decomposes the error into contributions from time discretization (Lemma 4.5), distributed learning dynamics (Lemma 4.6), local loss bounds (Lemma 4.7), and denoising score matching equivalence (Lemma 4.8). The use of Girsanov’s theorem (Lemma 4.9) to measure the discrepancy between the true and learned distributions appears correct and follows standard techniques in diffusion model analysis.
2. The proof of Lemma 4.6 follows a standard optimization analysis approach, bounding the expected gradient norm using assumptions on Lipschitz continuity and bounded variance. The decomposition of pruning-induced errors and stochastic gradient noise is logical, and the derivation steps are consistent with related works in distributed learning.
3. Lemma 4.8 shows that the local loss function used in training is equivalent to the theoretical loss function up to a constant. The derivation follows from the Gaussian noise assumption in the forward process, which is a well-established technique in score-based generative modeling.

---

> ### Author Rebuttal · Authors · 2025-03-31
>
> We thank the Reviewer ySb9 for the time and valuable feedback! We would try our best to address the comments one by one.
>
> **Response to “Essential References Not Discussed”:**
> We appreciate the reviewer’s insightful suggestion. While our work primarily focuses on theoretical contributions, we agree that incorporating discussions of experimental works in distributed diffusion models can further underscore the practical relevance of our analysis. In response, we have cited some works on the empirical performance of distributed diffusion models, such as Phoenix and FedDM. We believe these additions enhance the manuscript.
>
>
> **Response to “Other Weaknesses 1” ＆ “Other Comments Or Suggestions 1”:**
> We appreciate the reviewer’s comment on the balance between mathematical rigor and intuitive clarity. In response, we provide a few brief intuitive explanations. For the reverse SDE derivation, we now clarify that reversing the SDE involves “rewinding” the forward process by inverting the drift, which conceptually aligns with the idea of reconstructing the data distribution from noise. For the local loss bound (Lemma 4.7), it is directly derived based on the fact that the average gradient norm is non-negative. We hope these explanations address your concerns.
>
>
> **Response to “Other Weaknesses 2” ＆ “Other Comments Or Suggestions 2” ＆ “Questions For Authors 3”:**
> We appreciate the reviewer’s insightful suggestion. To make it clearer how the results in Remark 4.12 are established, we have added a detailed derivation in the Appendix. Here are the details:
> For $T\ge 1$, $\delta < 1$, $K \ge \log(1/\delta)$, if we set $\kappa = \Theta \left(\frac{T + \log(1/\delta)}{K}\right)$, then there obviously exists a sequence $\\{t _ k\\} _ {k=0}^K$ such that $\gamma_k \le \kappa \min \\{1, T-t_{k+1}\\}$.
> Then, if we set $K=\Theta \big(\frac{(d+M_{n,2})(T+\log (1/\delta))^2}{F(\theta_0)}\big)$, it holds
> $$
> \begin{equation}
> \\left\\{
> \begin{aligned}
> &\kappa^2 d K=\Theta \Big( \frac{d F(\theta_0)}{d+M_{n,2}} \Big)\lesssim F(\theta_0)\\\\
> &\kappa M_{n,2}=\Theta \Big( \frac{M_{n,2} F(\theta_0)}{(d+M_{n,2})(T+\log (1/\delta))} \Big)\lesssim F(\theta_0)\\\\
> &\kappa dT =\Theta \Big( \frac{dT F(\theta_0)}{(d+M_{n,2})(T+\log (1/\delta))} \Big)\lesssim F(\theta_0)
> \end{aligned}\\right.\notag
> \end{equation}
> $$
> If we set $T=\frac{1}{2}\log \big(\frac{d+M_{n,2}}{F(\theta_0)}\big)$, it holds that $(d+M_{n,2})e^{-2T}=F(\theta_0)$. Then we have $\kappa^2 d K+\kappa M_{n,2}+\kappa dT+(d+M_{n,2})e^{-2T}\lesssim F(\theta_0)$.
>
> Similarly, if we further control the learning rate to satisfy $\eta \le\\{\frac{F(\theta_0)\Gamma^*}{SRN(\sigma_1^2+\sigma_2^2)},\frac{F(\theta_0)\Gamma^*}{SRNw^2 L},\sqrt{\frac{F(\theta_0)(\Gamma^*)^2}{SRNL\sigma_1^2}}\\}$, we have $\frac{\eta SR w^2 LN}{\Gamma^*}+\frac{\eta SRN(\sigma_1^2+\sigma_2^2)}{\Gamma^*}+\frac{\eta^2 SRLN\sigma_1^2}{(\Gamma^*)^2}\lesssim F(\theta_0)$.
>
> These results complete the proof. And we hope these explanations address your concerns.
>
>
> **Response to “Questions For Authors 1-2”:** Thank you for these thoughtful questions. **For your concern about the constant $C_1$**, we believe that the current version is correct. In fact, $C_1$ is generated due to the equivalent denoising score matching, which we have discussed in Lemma 4.8. Therefore, this constant is not included in the bound of local loss $F_n(\theta_R)$. **For your concern about $\sigma_2$**, it is indeed equal to zero and can be omitted. Thank you for your careful consideration, we have simplified it in the revised version. **For your concern about $w^2$ and  $\\Gamma^{*}$**, the term $w^2$ reflects the extent of pruning applied to the distributed model. As pruning becomes more aggressive (i.e., fewer parameters are retained), this term increases, leading to a looser error bound. The parameter  $\\Gamma^*$ captures the frequency with which each model parameter is updated across the distributed workers. A small  $\\Gamma^*$ indicates that some parameters are rarely trained, which may lead to suboptimal updates and thus a larger bound. This motivates us to seek more effective pruning strategies that achieve a balance between resource availability and generation quality, which we leave for the future.
>
> If there are any further confusions/questions, we are happy to clarify and try to address them. Thank you again and your recognition means a lot for our work.

---

### Official Review · Reviewer_grcd · 2025-03-13

**Overall Recommendation:** 4

**Summary:**

In this work, the authors investigate the impact of distributed collaboration on diffusion model training in environments with heterogeneous computational resources and data availability. It establishes the first theoretical generation error bound for distributed diffusion models, demonstrating a linear relationship with the data dimension d and consistency with single-worker results. A novel privacy-preserving training mechanism is proposed, incorporating synchronized noise scheduling and local sparse training to enhance computational efficiency. The study further highlights the critical role of hyperparameter selection, such as learning rate and noise scheduling, in optimizing generation quality. The findings provide a theoretical foundation for deploying diffusion models in distributed and resource-constrained settings while ensuring robust performance.

**Claims And Evidence:**

This paper makes several key claims about the theoretical analysis of distributed diffusion model training, supported by rigorous theoretical derivations. The following is an assessment of the main claims and their supporting evidence:
1. The paper derives a generation error bound for distributed diffusion models under resource-constrained scenarios. The analysis builds upon Girsanov’s theorem to measure the difference between ideal and actual data distributions. It demonstrates that the error scales linearly with the data dimension d, consistent with single-worker results. The theoretical results are formulated through KL divergence-based analysis and validated with convergence proofs.
2. The error decomposition in Theorem 4.11 shows that the main contributor to generation errors stems from distributed training constraints rather than local model accuracy alone.
3. The paper systematically analyzes how hyperparameters (learning rate, noise scheduling, pruning strategies) affect model convergence and performance.

**Essential References Not Discussed:**

In this paper, the authors provide a thorough discussion of prior work, covering key contributions in diffusion model theory, distributed learning, and generative model optimization. It appropriately cites foundational studies on diffusion model error bounds (e.g., Benton et al., 2024; Chen et al., 2022), as well as relevant work on federated and distributed training (e.g., Li et al., 2024 on DistriFusion). Given the scope of the paper, there do not appear to be major missing references that are essential for understanding its contributions. The cited works sufficiently frame the theoretical advancements and position the study within the broader landscape of distributed diffusion models.

**Experimental Designs Or Analyses:**

Since this is a theoretical paper, the experimental design and analyses are included only in the appendix and primarily serve to illustrate the theoretical findings. Below is an evaluation of the soundness of these experiments:
The experimental design effectively illustrates the theoretical findings by simulating a distributed training scenario with different pruning strategies on CIFAR-10, SVHN, and Fashion-MNIST. The use of training loss, IS, and FID as evaluation metrics provides a reasonable assessment of model behavior under resource constraints.
While primarily a theoretical study, the experiments in the appendix support key claims and offer insights into the practical implications of distributed diffusion model training.

**Methods And Evaluation Criteria:**

The methods and evaluation criteria in this paper are well-aligned with its theoretical nature. The study develops a rigorous mathematical framework for analyzing distributed diffusion model training, leveraging KL divergence, Girsanov’s theorem, and stochastic gradient methods to derive a generation error bound. These methods are appropriate for assessing theoretical performance under resource-constrained and heterogeneous environments.
The key evaluation metric, KL divergence, effectively quantifies the difference between the idealized and actual data distributions in distributed training settings. Additionally, the study considers error decomposition due to factors like time discretization, sparse training, and distributed model aggregation, ensuring a comprehensive theoretical assessment.

**Other Comments Or Suggestions:**

The paper is well-structured and clearly written, but here are a few minor suggestions for improvement:
1. Some proofs, particularly in Lemma 4.6 and Theorem 4.11, could benefit from additional intuition or visual explanations to improve readability for a broader audience.
2. A few sentences in the introduction and conclusion could be slightly reworded for clarity and conciseness. For example, the phrase "This discrepancy in resources and data diversity challenges the assumption of accurate score function estimation foundational to single-worker models" could be made more direct.
3. Conduct a final proofreading pass to catch any minor grammatical errors or inconsistencies in notation, particularly in equations.
These are minor refinements, and the paper is already well-organized and rigorous in its theoretical contributions.

**Other Strengths And Weaknesses:**

The paper demonstrates strong originality by extending single-worker diffusion model theory to the distributed setting, providing the first known theoretical generation error bound under resource constraints. This is a significant contribution, as most prior theoretical analyses of diffusion models have focused on centralized training. The study also introduces a novel distributed training mechanism that incorporates privacy-preserving synchronized noise scheduling and sparse training via pruning, which aligns well with real-world constraints in federated and distributed learning.
The theoretical rigor is a key strength, as the paper provides clear derivations using KL divergence, Girsanov’s theorem, and stochastic gradient analysis to quantify the impact of distributed training on generation quality. The mathematical results are well-structured and contribute to a deeper understanding of error propagation in distributed diffusion models.
In terms of clarity, the paper is generally well-written, with precise definitions and clear explanations of theoretical results. However, some technical sections—particularly the proofs—could benefit from additional intuition or visual explanations to improve accessibility for a broader audience.
A potential consideration is that some assumptions like synchronized updates and bounded variance, though standard in theory, may not fully capture practical distributed learning challenges.

**Questions For Authors:**

1. Could the authors explain in detail how (15) is summed to obtain (17) in L.721? Is there a formula citation error here, i.e. (15) should be (16)?
2. If the initial samples of all workers are identically distributed, does that mean sigma_2=0 in remark 4.12? If so, then the corresponding bound of F_n(theta_R) can eliminate sigma_2.
3. As far as I know, a lot of work on single-node diffusion models requires the score function to be Lipschitz continuous with respect to the data, such as [1]. Does this paper use the same assumption?
[1] Chen H, Lee H, Lu J. Improved analysis of score-based generative modeling: User-friendly bounds under minimal smoothness assumptions[C]//International Conference on Machine Learning. PMLR, 2023: 4735-4763.

**Relation To Broader Scientific Literature:**

This work builds on existing research in diffusion models, distributed learning, and theoretical generative modeling, extending key findings to a distributed setting. It connects to prior work on diffusion model error bounds (Chen et al., 2022; Benton et al., 2024) by establishing the first known generation error bound for distributed diffusion models under resource constraints, aligning with single-worker results. Additionally, it draws from federated learning by incorporating sparse training and pruning strategies to optimize model performance in heterogeneous environments. The study also reinforces prior findings on hyperparameter selection in generative models (Song et al., 2020) by demonstrating that training dynamics, rather than local accuracy alone, drive generation quality in distributed training. By bridging the gap between diffusion model theory and distributed optimization, this paper provides a theoretical foundation for scalable, resource-efficient generative modeling.

**Theoretical Claims:**

The paper presents theoretical claims regarding the generation error bound for distributed diffusion models. These claims are primarily supported by mathematical proofs leveraging KL divergence, Girsanov’s theorem, and stochastic gradient analysis. Below is an assessment of the key proofs:
1. Time Discretization Error (Lemma 4.5): The proof applies Itô calculus to analyze the impact of discretized time steps on the reverse stochastic differential equation (SDE). The result extends previous single-worker error bounds to the distributed setting, ensuring consistency with established theoretical findings.
2. Distributed Learning Dynamics (Lemma 4.6): This lemma establishes the convergence rate of the gradient norm in a distributed training framework with pruning. The proof correctly derives an upper bound on the expected gradient norm over multiple training rounds, accounting for pruning-induced errors and stochastic noise. While the proof relies on the bounded variance assumption (Assumption 4.3), which may be restrictive under extreme data heterogeneity, this assumption is widely accepted in distributed learning and does not significantly impact the theoretical validity.
3. Distance Between Path Measures (Lemma 4.9): This proof uses Girsanov’s theorem to derive an upper bound on KL divergence, linking the idealized and practical training processes in a distributed setting. It extends the analysis from single-worker diffusion models to multi-worker collaboration while maintaining theoretical rigor.
4. Generation Error Bound (Theorem 4.11): The theorem quantifies the gap between the idealized data distribution and the actual learned distribution using KL divergence. The proof decomposes the overall generation error into distinct contributions from time discretization, distributed training dynamics, and early stopping effects, providing a comprehensive theoretical foundation for analyzing distributed diffusion models.
In summary, the proofs are well-structured, mathematically sound, and extend existing theoretical results to the distributed setting. While certain assumptions—such as bounded variance and synchronized updates—may not fully capture real-world constraints, they are reasonable within standard theoretical frameworks for distributed learning. Overall, the claims made in the paper are strongly supported by rigorous mathematical analysis.

---

> ### Author Rebuttal · Authors · 2025-03-31
>
> We thank the Reviewer grcd for the time and valuable feedback! We would try our best to address the comments one by one.
>
> **Response to “Other Weaknesses 1” ＆ “Other Comments Or Suggestions 1”:**
> We thank the reviewer for this constructive feedback. We agree that the proofs in Lemma 4.6 and Theorem 4.11 could benefit from additional intuition and visual aids. In response, we have enhanced the manuscript by adding more detailed explanations that clarify the logical steps. These revisions are intended to make the technical sections more accessible to a broader audience while preserving the rigor of our arguments. We believe these improvements will significantly enhance the readability and overall presentation of our work.
>
> **Response to “Other Weaknesses 2”:**
> We appreciate the reviewer’s observation. Indeed, while assumptions like synchronized updates and bounded variance are common in theoretical analyses, we recognize that these assumptions may be overly idealized when compared to the complexities of practical distributed learning scenarios. We adopted these assumptions to align our work with the established literature on distributed learning. In the revised version, we have included an expanded discussion on potential limitations arising from these assumptions, along with directions for future work to extend our framework to settings with asynchronous updates and more realistic variance conditions. We believe these clarifications strengthen the paper and provide a balanced view of both the theoretical guarantees and practical applicability.
>
> **Response to “Other Comments Or Suggestions 2-3”:**
> We appreciate the reviewer’s detailed suggestions. In response, we have reworded several sentences in the introduction and conclusion for improved clarity and conciseness. For example, we revised the phrase to: "The disparity in resources and data diversity undermines the accurate score function estimation assumed in single-worker models." Additionally, we conducted a thorough proofreading pass to address minor grammatical errors and ensure consistency in notation, particularly within the equations. We believe these refinements enhance the clarity and overall quality of the manuscript while maintaining its theoretical rigor.
>
> **Response to “Questions For Authors”:**
> **Regarding your first question**, we would like to thank you for your carefulness. There is indeed a citation error here, that is, (15) should be (16), which we have corrected in the revised version. We next explain how to get (17) from (16).
> By summing (16) from $s=1$ to $S$, from $n=1$ to $N$, and from $r=1$ to $R$, we can obtain the following inequality:
> $$
> \begin{align}
> \sum_{r=0}^{R-1}\sum_{n=1}^N \sum_{s=1}^S \mathbb{E} \parallel  \theta_{r,n,s-1}-\theta_{r} \parallel^2
> \le& 8\eta^2 S^2 L^2\sum_{r=0}^{R-1}\sum_{n=1}^N \sum_{s=1}^S \mathbb{E} \parallel  \theta_{r,n,s-1}-\theta_{r} \parallel^2+ 8\eta^2 S^3 N R(\sigma_1^2+\sigma_2^2)+8\eta^2 S^3 N\sum_{r=0}^{R-1}\mathbb{E} \parallel F(\theta_{r})\parallel^2+ 2w^2 RSN  \notag
> \end{align}
> $$
> By shifting term and dividing both sides of the inequality by $\Gamma^*$, we can obtain (17).
>
> **For your second question**, if the initial samples across all workers are identically distributed, then $\sigma_2^2$—which quantifies the discrepancy among workers—would be zero. Consequently, the bound on $F_n(\theta_R)$ simplifies by eliminating the $\sigma_2^2$ term. Thank you for your careful consideration, we have simplified this representation in the revised version. **For your concern on the third question**, we avoid using Lipschitz continuity assumption on the data due to its inherent limitations. A uniform Lipschitz condition can be overly restrictive, as the associated constant may scale with the data dimension—particularly when the distribution is approximately supported on a sub-manifold. Moreover, even employing a time-varying Lipschitz constant does not fully mitigate this issue. For example, Chen et al. (2023) assume Lipschitz smoothness at $t=0$ but still obtain a quadratic dependence on the data dimension $d$. Consequently, we choose not to rely on the Lipschitz assumption with respect to the data.
>
> If there are any further confusions/questions, we are happy to clarify and try to address them. Thank you again and your recognition means a lot for our work.

---

### Official Review · Reviewer_hsmD · 2025-03-14

**Overall Recommendation:** 4

**Summary:**

This theoretical paper analyzes the possibilities of distributed training of diffusion models The authors propose a new, privacy-preserving approach to distributed training of diffusion models and present a proof of the error bound. They further analyze hyperparameter adjustments to improve performance in this setting.

**Claims And Evidence:**

The authors claims are primarily backed up via proofs, which is appropriate given the theoretical focus of the paper.

**Essential References Not Discussed:**

To the best of my knowledge the authors did not miss any essential references.

**Experimental Designs Or Analyses:**

Outside of the supplementary material there are no traditional experimental designs. The supplementary material includes what is essentially an ablation study that further supports the authors' theoretical contributions.

**Methods And Evaluation Criteria:**

The method and evaluation criteria are appropriate for the paper's claims. I did appreciate the inclusion of experimental results in the supplementary materials. But it would have been beneficial to also include example model outputs to further characterize the generation error bounds.

**Other Comments Or Suggestions:**

No other comments or suggestions.

**Other Strengths And Weaknesses:**

The paper is very well written and argued. I appreciate the inclusion of Figure 1 to help intuitively explain the approach despite the authors theoretical focus.

**Questions For Authors:**

1. Can the authors share outputs of the approach and its ablation? If not, is Figure 1's illustration representative or generation or just the training process?

**Relation To Broader Scientific Literature:**

The authors do a good job of positioning this work in terms of practical diffusion model research, theoretical diffusion model research, and distributed learning and privacy-preserving research. I have no complaints about this aspect of the paper.

**Theoretical Claims:**

I checked the correctness of the proofs to the best of my ability. All proofs appeared to be well-structured.

---

> ### Author Rebuttal · Authors · 2025-04-01
>
> We thank the Reviewer hsmD for the time and valuable feedback! We would try our best to address the comments one by one.
>
> **In response to the concern about the outputs of diffusion models**, we have provided additional Figures 4-6 in Appendix E.3, which can also be found at the anonymous link: https://anonymous.4open.science/r/Diffusion-0D86/
> Specifically, we randomly selected 30 noise instances following Gaussian distribution, and then generated 30 samples from them. This procedure was applied to each pruning setting of both pruning techniques across all datasets.
>
> The generated samples shown in Figures 4-6 further reveal that pruning influences generation quality, particularly on complex datasets. For CIFAR-10 and SVHN, the full model produces images with more details and consistent color distribution, whereas aggressive pruning leads to noticeable degradations—such as blurred features, increased noise, and color distortions. Top‑k pruning preserves critical parameters more effectively, yielding images that are more closed to those produced from the full model, particularly under moderate pruning conditions. For simpler datasets like Fashion‑MNIST, the impact of pruning is less severe; in fact, higher pruning levels sometimes result in better images due to a regularization effect that eliminates redundant details and prevents overfitting.
>
> **For the concern about the Figure 1**, it serves as an illustration of distributed diffusion model training with pruning. Through Figure 1, we aim to help readers comprehend the training process as well as key aspects of the theoretical analysis, such as time discretization error, distributed training dynamics, and the impact of early stopping.
>
> If there are any further confusions/questions, we are happy to clarify and try to address them. Thank you again and your recognition means a lot for our work.

---

> > ### Comment · Reviewer_hsmD · 2025-04-02
> >
> > Thanks to the authors for the additional outputs and clarification Figure 1. These are both very much appreciated. Given that I am already advocating for acceptance I won't be changing my review recommendation.

---

> > > ### Author Response · Authors · 2025-04-07
> > >
> > > Once again, thank you for your invaluable feedback. We sincerely appreciate your support and approval of our work!

---

### Decision · Program_Chairs · 2025-05-01

**Decision:**

Accept (poster)

**Comment:**

This paper considers training diffusion models in resource-constrained distributed settings, whereby it provides a generation error bound which can be controlled based on the choice of certain hyperparameters, depending on the training dynamics. Its techniques draw on a variety of tools from probability theory, and the reviewers are all in agreement that this work presents important theoretical contributions to our understanding of distributed training for diffusion models. The paper would therefore be a welcome addition to the conference.